biogeography/environmental science/ecology

historical ecology, social-ecological systems, novel ecosystems, historical geography, charcoal production, landscape transformation

**Author for correspondence:**
Alexandro Solórzano
e-mail: alexandrosol@gmail.com

# Land use and social-ecological legacies of Rio de Janeiro's Atlantic urban forests: from charcoal production to novel ecosystems

Alexandro Solórzano, Ana Brasil-Machado and
Rogério Ribeiro de Oliveira

Biogeography and Historical Ecology Laboratory (LaBEH), Department of Geography and Environment, Pontifícia Universidade Católica do Rio de Janeiro, Rua Marquês de São Vicente 225, Rio de Janeiro 22451-900, Brazil

AS, 0000-0001-7562-0720

Historical ecology is an important tool in deciphering human–environment interactions imprinted on landscapes throughout time. However, gaps of knowledge still remain regarding the land use legacies hidden in the current Atlantic Forest landscape; and also regarding how this information can help management of the remaining forest cover. The social-ecological systems framework was applied to understand charcoal production in the urban forests of Rio de Janeiro, from the nineteenth to mid-twentieth century, and their current social-ecological legacies. Charcoal production carried out by former enslaved populations, allowed for rapid forest regeneration. Forest thinning instead of forest felling was carried out by small groups in these urban remnant forests, sparing large native trees and facilitating natural regeneration. Currently, more than one thousand former charcoal production sites are accounted for hidden underneath the forest cover. The forest landscape of today is a result of novel forest successional trajectories that recovered structural and functional attributes of the forest ecosystem. However, this came at the cost of social invisibility and marginalization of these populations. The management practices of charcoal production dispersed in the landscape is one of Rio de Janeiro's most important, albeit hidden, land use legacies. Currently, the forested landscape is comprised of regenerated forests, both structurally and functionally sound, though with significant changes in species composition including the introduction of

exotic species throughout recent centuries. These urban forests are today a complex mosaic of novel ecosystems, with rich biocultural diversity, and together with managed lands and well conserved forest tracts, provide not only livelihood and sustenance for forest dwelling families, but also important ecosystem services for the entire population of Rio de Janeiro. We believe that these concepts and frameworks can offer practical solutions for urban forest management, taking into account the biocultural diversity of Rio de Janeiro, increasing awareness of sustainability and promoting food security.

# 1. Introduction

Currently, it is widely accepted that it is impossible to understand society without nature, and nature without society. This statement becomes more evident with a broader understanding that we currently live in the Anthropocene—the human epoch [1]. Tropical landscapes, and the ecosystems they contain, are continually in transformation, being directly influenced by natural climatic, geological and evolutionary changes. In addition to these natural constraints, anthropogenic pressures have increasingly influenced ecosystems, transforming the landscape over the past few centuries. Thus, the biophysical conditions of the landscape in which species adapt to and interact have been modified in their most basic structure and composition, changing the evolutionary trajectories of biological communities and affecting ecosystem processes in their biotic and abiotic components. This could potentially lead to changes never seen before [2,3].

## 1.1. Social-ecological systems

In a globalized society there are no ecosystems without humans nor people that live without the benefits of ecosystem functions [4]. These two components—social systems and ecological systems—are in fact interconnected and interdependent, thus functioning as social-ecological systems. Social-ecological systems is an important theoretical and methodological framework used in the emerging field of sustainability science; often termed as coupled human–environment systems [5,6]. This coupled system is described as complex and adaptive consisting of a biogeophysical unit, social actors and institutions, including their rules and decision making [7]. The social-ecological systems framework has been applied to natural resource management, shifting towards a stewardship of interdependent social-ecological systems paradigm instead of the traditional one resource at a time management style [4]. Folke & Berkes [8] used the social-ecological systems framework to broaden understanding of how social-ecological resilience is built into local resource management systems, and to find ways to match institution dynamics with ecosystems dynamics in order to achieve mutual social-ecological resilience.[1]

Social-ecological system research has also been used in social research today, as the results of past human–environment interactions have implications for the common future of humanity. Thus, historical ecology, environmental history and historical geography are positioned as a tool key to the resolution of concrete problems and the proposition of public policies related to social-ecological systems [9]. Understanding how societies used natural resources in the past, producing landscapes imprinted with our history, has become a very important field of scientific inquiry in recent decades. Without knowledge about how the landscape transformation process took place (including specific actors involved, detailed land use and management techniques and spatial-temporal scale of the activities impact) management decisions cannot be made regarding its land use, resource management and biodiversity conservation.

## 1.2. Historical coupled human–environment investigation

Some sub-fields and disciplinary traditions have contributed to constructing an approach that seeks to overcome the separation between nature and society [10]. Among them, we highlight three contributions: historical ecology, environmental history and historical geography. The physical evidences of human interaction with ecosystems is etched in the landscape, forming complex mosaics that must be read for in depth interpretation [11,12]. Therefore, these three sub-disciplines provide important tools to understand the 'dialectical relations of human acts and acts of nature made

---

[1]Resilience reflects the capacity of a system to buffer and survive disturbance, and also to deal with change and develop. Resilience allows for the conservation of information, knowledge and experience (the memory of the system) [8].

manifest in the landscape' [13, p. 9], shedding light on how land use and forest management practices in the past have shaped todays ecosystems [14].

Historical ecology is defined as 'the study of past ecosystems by charting the change in landscapes over time' [13, p. 6]. As a research programme it focuses on interactions between environments and societies over time and the consequence of these interactions in the formation of current and past cultures and landscapes [15], considering the social-ecological results of this interaction [16,17]. Historical ecology commonly considers human activity as one of the factors that influence ecosystems [18] and can inform the reference conditions necessary for landscape reconstruction [15]. Balée [15] has even equated restoration ecology with applied historical ecology, since knowledge of past species composition together with detailed past land use information and traditional knowledge are essential to restore past landscapes.

Geography is a discipline that has a lot to contribute to the investigation of human–environment interactions because it offers a conceptual contribution to address the relationship between society and nature. It can contribute to the cross-scaling understanding of biophysical factors and of power relations [19]. The specificity of historical geography resides in the relationship between spatial aspects and the physical marks or characteristics that different societies have imprinted on landscapes according to their specific cultural conceptions or forms of territorial manifestation [19]. Another important difference is the central role of fieldwork and observation of these marks in the landscape [20]. Environmental history, in its turn, emerged from political and moral concerns and developed as an academic enterprise that seeks to understand how humans were affected by their natural environments and how they affected those over time [11].

Despite the differences between historical ecology, environmental history and historical geography, these three fields offer fundamental tools for the historical understanding of the relationships between cultures and nature. In addition to the temporal approach, such fields are open to interdisciplinarity, emphasize the landscape as a fundamental category of analysis and operate from an evidential paradigm [21], observing different signs, such as documents, landscape forms and species assemblages. In this paper, we draw from the conceptual and methodological background of these three historical-ecological fields of research. From the intellectual influence [9,13–15,22], to the fieldwork-based investigation [20,23] and documental research [24,25], we grounded our evidence based findings and applied the collected material on the social-ecological systems framework in order to gain further insights on landscape level change and its consequences for current management practice in the urban Atlantic Forest remnants of Rio de Janeiro. This type of historically grounded social-ecological system analysis is a novelty for Atlantic Forest and urban ecosystems research and management.

## 1.3. Novel ecosystems and biocultural diversity

From the perspective of applied historical ecology, concepts have emerged as important tools to support the decision-making process, especially when considering the complexity of the social-ecological system and the limited resources to solve real-world problems. After human activities are abandoned, ecosystems go through different successional trajectories and self-regulation processes, giving rise to novel species configurations with their own ecological dynamics. Furthermore, novel or emerging ecosystems have composition and relative abundance patterns not previously seen in a given biome. They have arisen from anthropogenic environmental changes, changes in land use, introduction of exotic species or a combination of these factors [26]. These ecosystems are the result of human activity, that do not depend on their management for maintenance [27]. In summary, 'the essential characteristics of novel ecosystems, which distinguish them from unchanged (historical) systems, are: (i) a change in composition, structure or function; (ii) thresholds in these attributes—currently irreversible; (iii) persistence or self-organization' [28, p. 192]. Finally, these ecosystems can be considered as an intermediate form in a gradient ranging from 'natural' or 'wild' ecosystems, i.e. with few traces of human presence—to fully managed or man-altered systems.

There is an ongoing debate whether the novel ecosystem conceptual framework is useful for restoration ecologists and managers or a nuisance and inconvenient proposal [29,30]. Some embrace the flexibility offered by this framework [26,27,29] while others see it as a shift towards the abandonment of traditional strategies [30,31]. To better understand the debate involving this novel concept one must recognize that all ecosystem conservation and restoration decisions are socially constructed. Therefore, management of novel ecosystems should be contextualized within an social-ecological systems framework, being evaluated for its capacity to meet not only ecological indicators but also social and economic objectives [31, p. 116]. Backstrom et al. [31] proposed that 'analysing novelty within a decision context, against a range of ecological, social and economic management objectives, will be an effective way to reconcile conflicting stances on novel ecosystems'. In this sense, a growing number of conservation scientists and

practitioners are emphasizing the importance of novel ecosystems as a conservation framework, managing for ecological novelty that supports biodiversity and ecosystem service provision [32].

It is our understanding that the novel ecosystem concept has clear intellectual links with another novel conceptual framework: biocultural diversity. Both of these conceptual frameworks are clear attempts to frame humans and the environment in a non-dichotomous, 'people as part of nature' mode of investigation. Biocultural diversity, defined as the diversity of life in all its manifestations (i.e. biological, cultural and linguistic), has come to be understood as inter-related within complex and adaptive social-ecological systems [33]. Therefore, biocultural approaches hold great social-ecological transformative potential, by understanding the connections between cultural and biological diversity, human well-being, and social justice and by assisting the formulation of culturally relevant policies [34]. Most commonly, biocultural approaches are applied to investigate human–environment relationships with a consideration of cross-scale spatial-temporal interactions, using landscape both as a spatial unit and integrated concept [35]. Therefore, the biocultural approach, not only has links to the novel ecosystem and social-ecological system conceptual frameworks, but has theoretical and methodological ties to the above mentioned historical-ecological fields of research.

## 1.4. Human transformation of the Atlantic Forest

Ecosystem structure, on a wide scale, is predominantly explained by the variation in the physical environment, just as its current structure is strongly influenced by historical processes [36]. Our approach originates from the principle that, on a landscape scale, what we consider 'natural' today may in fact represent a system actively managed for centuries by past populations. The Atlantic Forest is a biome that covers almost the entire Brazilian coastline, and it extends into the interior up to 1000 km in some regions. Native South American, European and African populations occupied the forest for thousands of years, transforming and modifying it with their actions. These populations overlapped each other to varying degrees, often leaving visible traces, especially after the sixteenth century [37]. The remnant Atlantic Forest is comprised of native biodiversity that survived centuries of intense landscape transformation, together with biocultural diversity interspersed in the forest landscape including non-native species assemblages, novel ecosystems, and forest dwelling livelihoods and local ecological knowledge (such as banana and persimmon fruit extraction).

Historically, the proximity of the mountains to the city of Rio de Janeiro was responsible for transforming the Atlantic Forest covered slopes into areas for coffee plantations in the nineteenth century. Forest cover was also converted into pastures for horses used in urban transport and mainly as a source of firewood and eventually becoming a centre for the manufacture of charcoal. The demand for this energy source (charcoal) was considerable for the rising metropolis, becoming the main energy matrix for the city. Charcoal was used in factories, domestic use and in civil construction. Forges spread throughout the city, using charcoal as its basic input. They manufactured articles such as axes, hoes, sickles, car wheel rims and horseshoes, used for the hulls of the muar and equine troops. In Brazil, practically until the nineteenth century, a millenary process of direct reduction of ore (that is, by oxygen removal) was performed in small ovens using charcoal in the process. This charcoal production by use of forest biomass caused numerous modifications of the Atlantic Forest; however, many of the forest attributes were regenerated through natural ecological succession. The successional trajectory of the natural regeneration, in many parts of the forest, consists of novel species assemblages, i.e. novel ecosystems.

Although the Atlantic Forest is researched by several fields of scientific investigation, important knowledge gaps remain regarding its past use and dynamics. Furthermore, when carried out, research on the history of the Atlantic Forest is interested in a 'macro history' that considers major events such as coffee planting, deforestation, the water crisis and reforestation, using sources such as written documents and iconography. Our interest is to analyse the landscape transformations that occurred in the urban forests of Rio de Janeiro, considering the forms and practices spatially and temporally situated from the traces present in the forest today and taking the landscape in itself as a historical document [11]. Our main research question is: how can historically and geographically informed environmental investigation coupled with cross-disciplinary conceptual frameworks (such as social-ecological systems, novel ecosystems, biocultural diversity) help understand current landscape dynamics and inform decision-making? We also answer the following specific questions: (i) how did the charcoal workers transform the forest landscape? (ii) how did the vegetation structure, composition and functionality respond after charcoal production was abandoned? and (iii) how do the social-ecological systems and novel ecosystem frameworks improve the understanding of current social-ecological dynamics?

The main goal of this paper is to investigate the practices of the charcoal workers and the legacies derived from their work, since the social invisibility at that time has now become an invisibility of

their memory, their legacies in the forest and their crucial participation in the supply of energy for the city of Rio de Janeiro. Besides the 'detective work' in the landscape and the use of other documents, which are informed by historical geography, environmental history and historical ecology, we apply the social-ecological systems framework to gain insight and better understanding of the historical interconnectedness of charcoal producers with the forest, and to the underlying social-ecological legacies and biocultural diversity found in the contemporary urban forests of Rio de Janeiro, information without which it is not possible to carry out an adequate landscape management.

# 2. Material and methods

## 2.1. Study site

Our study was conducted in two large urban forest remnants of the Atlantic Forest biome, in the city of Rio de Janeiro: Pedra Branca and Tijuca. The city of Rio de Janeiro is the second largest urban city in Brazil and one of the largest economic and cultural centres in Latin America. In its 1204 km$^2$ total territorial area, 600 km$^2$ are urbanized, and with a population of 6 781 903 residents [38]. It has a peculiar geography being located between the coastal mountains and Guanabara Bay. The city's territory is composed of large areas, divided by two large massifs. On the city east side is the Tijuca Massif, and on the west, the Pedra Branca Massif. The two together have an area of approximately 280 km$^2$. The presence of these mountains generates a particular and uneven distribution of the city, composing an extremely diverse landscape. In this way, the presence of forests, beaches and mountain massifs contribute to the configuration and identity of the city, that also provide important ecosystem services.[2] The juxtaposition of natural elements and the urban matrix is a hallmark of the city, producing a strong image with global recognition. This harmonious mixture led to UNESCO naming Rio, in 2012, the title of World Heritage in the Cultural Landscape category [40]. We intend in this work to show that the landscape of Rio de Janeiro is configured as a hybrid between natural and manmade, filled with biocultural elements that integrates a complex social-ecological system.

In the last 500 years, Brazilian ecosystems have undergone major transformations, the Atlantic Forest being one of the biomes that suffered most from these changes. The social-ecological legacy imprinted on the Atlantic Forest landscape is derived from economic activities such as monocultures of sugar cane and coffee. The current configuration of this biome comprises extensive deforested areas, bare hillsides, silted rivers and forest remnants practically restricted to sloping and difficult to access land, in the form of secondary forests of different ages and successional trajectories [41,42]. The coastal massifs of Rio de Janeiro, that harbour the Tijuca and Pedra Branca remnant forests, were the scene of an extensive process of forest resources usage by slave populations and freemen for the production of charcoal as a means of subsistence. This work, largely invisible to society, allowed the establishment of these marginalized populations within the forest forming small *quilombola*[3] populations [47]. The peak of production occurred from the early nineteenth to mid-twentieth century, after the decline of monocultures and the abolition of slave labour.

## 2.2. Data collection and analysis

This paper in part synthesizes past studies conducted by two of the authors in the last decade, mostly published in Portuguese, and that can reach a new audience. We further apply the social-ecological systems framework proposed by Ostrom [48,49], a novelty for historical-ecological investigation of the Atlantic Forest, to better understand the interconnectedness of charcoal producers with the forest, the social-ecological legacies in the current landscape and potential solutions for present day forest management. Here we describe the main methods used for collecting historical, archaeological, vegetational and spatial data related to this social-ecological process of charcoal production and its consequences on landscape transformation, including changes on forest structure and composition.

---

[2]Ecosystem services are defined by the Intergovernmental Science-Policy Platform on Biodiversity and Ecosystem Services (IPBES) as 'the benefits people obtain from ecosystems' [39, p. 1]. In other words, ecosystem functions important for both human well-being and the rest of nature.

[3]*Quilombola* is the generic term to refer to black individuals participating in *quilombos*. These, in turn, were identified as African descendent rural isolated communities living off extractivism often originated by escaped or freed enslaved people. However, such communities have diverse origins, they could be located near farms and urban centres and they could maintain more or less intense relations with various sectors of society through the commercialization of their agriculture surplus, for example [43–46].

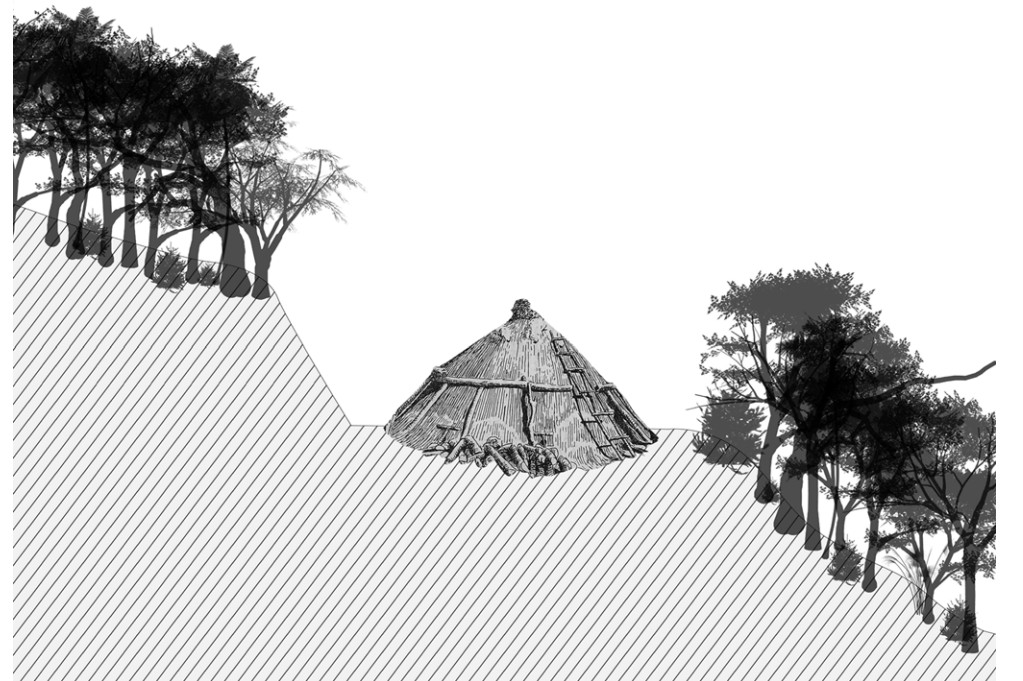

**Figure 1.** Scheme of a charcoal kiln carved into the forested slope of the Pedra Branca and Tijuca Massifs, Rio de Janeiro. Authorship: Isabel Machline.

Our historical-ecological mode of investigation is closely associated to the retrospective studies tradition in historical geography [20,50] and to the 'landscape-as-document detective-work' approach of environmental history and historical ecology [23,25,51]. The historical investigation was carried out using primary written sources (e.g. production reports, administrative inventories) and iconography (the study of photographs, prints and old paintings aimed at understanding landscape transformations), in addition to relevant secondary sources [52]. Bibliographic references of historical charcoal production in Latin America are scarce. We highlight the works of Thiéblot [53] for the state of São Paulo and García-Montiel [54] for Costa Rica. However, there is lack of documental information on charcoal production in the forests of southeast Brazil. Specifically for the charcoal manufacturing process, we found only generic references with a focus on metallurgy from the sixteenth to nineteenth century, such as the work of Landgraf *et al.* [55], or from the beginning of the twentieth century [56].

The present study was carried out in the two massifs in Rio de Janeiro city (Tijuca and Pedra Branca), that historically have been the open air laboratories for numerous ecological, social-ecological and historical-ecological 'detective-work' investigations by the authors in the past 20 years. Concerning sampling techniques, we employed field methods from historical ecology and geography to obtain spatial data on the location in the current forest remnants of important landscape legacies. Systematic and random field explorations were carried out in order to locate and georeference old charcoal kilns (for more detail see [52,53]; see also figure 1 for depiction of a charcoal kiln in the forest slope). These charcoal productions sites are recognizable in the field by the location of the plateaus that are dug into the hillside, and by the evidence of dark earth (anthropogenic soil) with charcoal fragments [57–59]. Together with the previous charcoal production sites we also georeferenced the mark of ruins of old houses, farms, water catchment systems, bridges and old roads [58]. We registered the location of species with cultural value, such as exotic fruit trees (e.g. jackfruit and banana) and sacred religious species (species of *Ficus* sp. genus) [52,58]. All of these important landscape legacies were marked using GPS equipment suitable for use under dense forest cover. In order to produce the spatialized information regarding the landscape legacies, we used ArcGis 10.2.1 software and developed the maps using the UTM 23S projection and WGS 84 Datum [58].

Throughout the past decade vegetation structure and composition inventories were carried out in different sites with evidence of charcoal production in the past, in both remnant forest areas, in order to understand how natural regeneration of the forest occurred and what successional pathway it went. We used field plots of $10 \times 10$ m or $5 \times 20$ m, depending on site characteristics, geographical aspects and charcoal production site distribution (for more detail see [60,61]).

The method for calculating the energy conversion of the produced charcoal is found in Oliveira *et al*. [57] and Oliveira & Fraga [62]. For the evaluation of the return time of nutrients exported from the forest to the city by wood extraction (in the form of charcoal), the litter deposited on the area explored in the charcoal clearing was considered. The return time evaluation of these nutrients to the system was made by dividing the total nitrogen, phosphorus and potassium exported in the wood by the inputs of these nutrients, either by atmospheric or by litter fall (details in [62]).

For the application of the social-ecological systems framework, we employed Ostrom's methodological approach [48,49] also known as the social-ecological system robustness framework [5]. The framework is based on eight first-tier core sub-systems defining the interactions (I) between four multi-linked subsystems which are resource units (RU), resource system (RS), governance system (GS) that include all of the decision-making processes (governmental and community level) and resource users (U). All of these variables deliver specific outcomes (O) and interact with the social, economic and political settings (S) and with their related ecosystems (ECO) [63]. Each first-tier variable is consequently sub-divided into second-tier variables that provide greater detailing for the interconnections between the first-tier variables (for complete list of second-tier variables see table 3).

# 3. Results and discussion

## 3.1. Land use history and charcoal production

Charcoal was the energy matrix of the city of Rio de Janeiro from the nineteenth century to mid-twentieth century. This form of energy was used in industry, for locomotives, in iron manufacturing, for domestic use, all of which enabled urban sprawl [47]. It was a highly complex social-ecological system, connecting city and forest, energy and human work in Rio's social-ecological metabolism [47].

A limited amount of written documentation exists about charcoal production prior to the twentieth century. References of charcoal producers, the main agents of this process, are even more scarce. In nineteenth century newspapers, the few mentions of these charcoal workers (*carvoeiros*) are coated with racial discrimination and prejudice of social status [64]. It should be noted that this social prejudice and invisibility originates from the fact that these charcoal workers, even after the abolition of slavery in Brazil, were possibly ex-slaves and *quilombolas*. In addition, their work was located in the mountains and forests on the fringes of this booming city [59].

Socially and spatially marginalized,[4] without access to land and the means of production, the charcoal activity demanded very few initial inputs (of tools) and guaranteed relative independence. Owing to the sparse documentation, the activity of the charcoal workers must be understood through the remnants of their material culture, signs and traces of which are visible in the current landscape. These include landmarks such as ruins and plateaus, tools and objects and the structure and composition of the current vegetation.

Borrowing a concept from Brazilian Geographer Paulo Cesar da Costa Gomes, such evidence constitutes today a 'geographical frame' (*quadro geográfico*), referring to geography as a way of structuring thought, as a graphic analysis in which we must consider different components (including elements originating at different times, such we propose here) and as a composition on which the spatial relationships between such components should be analysed [65]. This frame allows us to think about the relationships between forms and also actions and values in order to understand the energy supply of the city of Rio de Janeiro, whether on the scale of the forest landscape or on the scale of the city as a whole.

From the landscape point of view, the historic charcoal kiln remnants are recognizable underneath the forest canopy only by their platforms of levelled terrain. The charcoal workers manually excavated the slopes forming plateaus for the *in locus* charcoal production (see diagram in figure 1). A second important evidence of charcoal production is, together with the plateaus, the remains of the charcoal present in the soil, characterized as anthropogenic dark earth [66]. The average size of the charcoal kilns is estimated as 5.5 m diameter with a height of 3.3 m presenting a volume of 26.13 m$^3$ [57].

Patzlaff *et al*. [67] propose that the area effectively exploited by the charcoal producers had the approximate shape of a semicircle with a 60 m radius (representing an area of 0.5 ha), with the charcoal production as the centre. The area to be explored would be upslope of the charcoal plateau, using gravity

---

[4]The charcoal workers, already coming from a marginalized social condition (as ex-enslaved people), became even more invisible as their work was done in a remote setting, hidden under forest cover. Charcoal work was considered as one of the socially lowest type of work, owing to its rustic and dangerous nature, with dark coal staining their clothes and skin. This added to their already marginalized condition.

to facilitate down-slope timber transportation [60]. Solórzano *et al.* [58] calculated a 100 m radius centred on the charcoal plateau, referring to the area of influence of the charcoal production sites. This area of influence was used to understand the spatial association of charcoal site with the current distribution of exotic fruit tress (such as the jackfruit) in the forest. When the available trees were far from the plateau, the charcoal workers would build another and restart work. The scenery of hundreds of plateaus distributed along the slopes is what is found inside the current forest of the Pedra Branca and Tijuca Massifs [60]. However, large ancestral trees were preserved by the charcoal workers for two reasons: (i) the cultural value of large diameter trees (such as *Cariniana legalis*) as a form of respect of its ancestral legacy, or for religious beliefs of both Christian and African based religions that valued large fig trees (*Ficus* sp.) owing to their religious meaning and representation[5]; and (ii) for practical reasons as felling and transporting large trees would require a great expenditure of labour. Taking this cultural practice into account and excluding individuals with a diameter greater than 40 cm, Sales *et al.* [60] estimated an average supply of 260.4 m$^3$ firewood ha$^{-1}$ for forest tracts at different successional stages. The preservation of larger trees facilitates ecological succession and constitutes a cultural practice and labour technique that reduces overexertion. The main result of this was selective logging instead of clear cutting for charcoal production, which would facilitate and accelerate natural regeneration in charcoal production sites. This small detail, driven by cultural values and labour techniques, was an important component of the workers' management planning that led to a decrease of biomass recovery time. This is the first step in understanding this invisible and artisanal practice as being ecologically less impactful (considering its landscape legacies), compared to other land uses of the period, such as coffee monoculture.

## 3.2. Spatial distribution of vestiges

At present, 1176 charcoal kilns have been found and mapped in the Pedra Branca Massif and 346 in the Tijuca Massif (see map in figure 2), summing a total of 1522 inventoried sites in both massifs. Through extrapolation[6] we estimate that at least 3000 more charcoal production sites can be found in the urban remnant forests of Rio de Janeiro. The same forest space is shared by ruins (foundations) of old houses and coffee farms, water catchment systems, ancient bridges and stone paths. In the Pedra Branca forest 104 ruins were found and in the Tijuca Massif 107 ruins were found. It is possible to establish a relationship between the ruins and the charcoal labourers' activities. It is very likely that the exploration being carried out in remote locations, on the slopes of these mountains, required the charcoal workers to live close to the production sites. Abandoned foundations made of stones are commonly found close to the charcoal production sites [47]. Figure 3 shows the altitudinal distribution of the vestiges (charcoal and ruins) found in the Pedra Branca Massif: the majority of charcoal sites and ruins are located between 200 and 300 m altitude; only 30% are above 300 m; and only 33.6% of the charcoal kilns and 35.4% of the ruins are distributed at less than 200 m of altitude. The altitudinal distribution pattern of charcoal sites and ruins in the Tijuca Massif is similar to the one found at Pedra Branca, with two-thirds of the vestiges concentrated between 100 and 400 m (figure 4). This fact shows that the charcoal exploration palaeo-territory[7] was preferably located in steep and elevated areas, distant from level terrain and unsuitable for agriculture. This landscape occupation pattern is suggestive that the charcoal workers did not compete with farmers, who had their plantations and houses located in the piedmont region of the massifs. It also suggests that this exploitation was not being carried out at the behest of these wealthy landowners or by socially accepted farmers. It is very likely that they did not want to be seen or discovered. After the abolition of slavery, this occupation pattern of elevated forested areas was maintained, suggesting that it was a parallel and marginal occupation to the farmers and land owners.

[5]In the Judeo-Christian tradition the tradition the fig tree appears in many places in the Bible. In the Gospel of St. Mark (chapter XX) Jesus dries out a fig tree that did not bear fruit. This image permeates the collective unconscious of rural populations and has turned the fig tree into a tree with connections to the divine. In the Brazilian Afro-descendant tradition the fig tree takes the place of an African species to represent an *Orixá* called *Iroko* [68], the first tree to be planted in ancient times [69].

[6]We carried out a simple extrapolation, considering a 100 m area of influence around each charcoal site inventoried, in the Tijuca and Pedra Branca Massifs. We then extrapolated the value we found to the entire area of both massifs. This averages out the locations with a probability of concentrating a higher amount of charcoal sites (elevation, slope, proximity to old farms etc.) with the locations with a probability of having little to no charcoal sites (remoteness, difficulty of access etc.).

[7]Palaeo-territory is a concept that expresses the spatialization of past human-ecosystem interactions that left distinct ecological legacies in the landscape. Therefore, it is the physical and biological evidence of ecosystems being used by past populations in search of their subsistence, or economic activities, e.g. charchoal palaeo-territories, coffee palaeo-territories, *caiçara* slash-and-burn palaeo-territories [23].

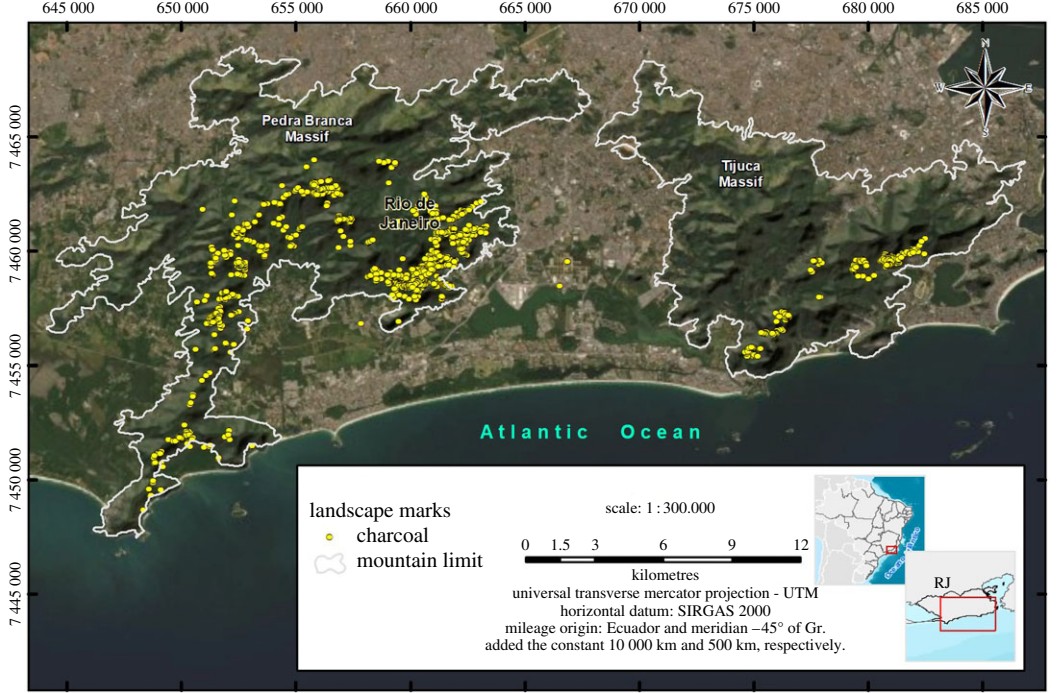

**Figure 2.** Map of the vestiges of charcoal production sites from the nineteenth and twentieth centuries, in the Pedra Branca (left) and Tijuca (right) Massifs, Rio de Janeiro, Brazil. Author: Maria Luciene da Silva.

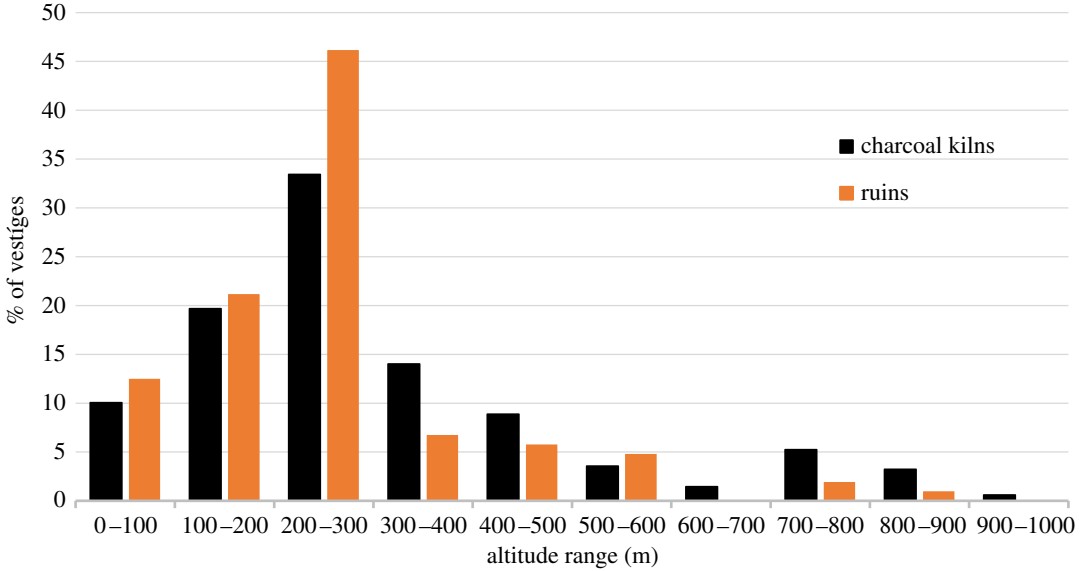

**Figure 3.** Percentage of the vestiges (charcoal kilns and ruins) altitudinal distribution in the Pedra Branca Massif.

## 3.3. Vegetation analysis

Tropical forests have a very effective regeneration capacity. However, there are many factors that influence this regeneration, such as disturbance intensity, regeneration time, spreading agents of propagules (wind, animals, water), surrounding vegetation and physical factors (slope orientation, soil humidity, insolation) [70,71]. Despite the significant forest area used by the charcoal workers, the forest has returned primarily to ecological succession that occurred after the selective logging of trees. Today, these charcoal kilns and ruins are practically unrecognizable, completely covered by a dense forest vegetation. Of the total charcoal production sites found in the Pedra Branca Massif (1172 charcoal plateaus), only 27 (2.3%) were in grassland areas (pastures) and 12 (1.0%) in banana plantations. The remaining 96.7% were in forested areas. Of the 346 charcoal sites found in the Tijuca

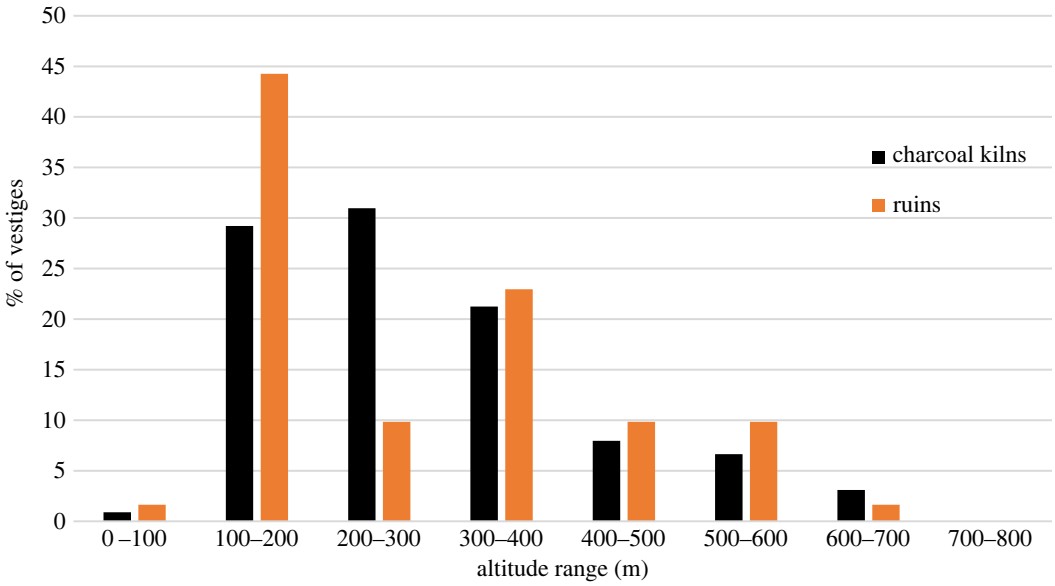

**Figure 4.** Percentage of the vestiges (charcoal kilns and ruins) altitudinal distribution in the Tijuca Massif.

Massif, all of them were under dense forest cover. The regrowth of stumps may have played an important role in the return of the forest: some species that were cut for charcoal production are able to regrow and, because they lose apical dominance, they created bifurcations in the regrowth process (figure 5) [72,73]. When the trees were cut, both the stump and the root system were preserved, which favoured resprouting with numerous ramifications from the point of cutting. These species have in common the fact that the bifurcation is located at a height of 0.5 to 1 m from the ground (knee and hip height), indicating the variable height where the axe cut the tree [72]. This pattern is relatively common in areas that were deforested and constitutes a macro vestige (visible aspects of material culture) of past use of the forest. The resprouted tree stumps compose one of the vegetation indicators of past forest use alongside native species with anomalous distributions and persistence and dominance of previously introduced exotic species, especially fruit trees and ritualistic herbs [72].

Species introduction in these forests provided a source of novelty and vegetation change. In the Pedra Branca and Tijuca Massif, jackfruit (*Artocarpus heterophyllus* Lam.), a fruit species native of southeast Asia, was introduced to feed the slave populations in the farms and it was also consumed by the charcoal workers owing to its abundance and low cost [17,58]. Jackfruit is rich in carbohydrate and protein and was introduced in farms and transported to the charcoal production sites in the forest, germinating from the uneaten remains discarded in the forest. Currently over 51.5% of the jackfruit found in the southern facing slopes of the Tijuca Massif are spatially associated with charcoal production sites (considering a 100 m influence area around the sites) [58]. Almost 90% of jackfruit populations overlap with areas with traces of human use and occupation of the forest (summing the charcoal sites with farm ruins, trails, ancient road networks and forest edge), revealing an intimate relationship of the *carioca*[8] culture with the jackfruit [17,58]. On the forest edge, along abandoned farms (figure 6a) and among old charcoal production sites (figure 6b), the jackfruit comprises up to 70% of relative density and dominance of these forest tracts. Thus, the jackfruit represents a legacy of human presence in the forest, as a biocultural element that makes up the social-ecological system of the Tijuca Forest, composing what has been classified as jackfruit novel ecosystems (figure 6b). Other fruit species such as mango (*Mangifera indica* L.), rose-apple (*Syzigium jambos* (L.) Alston), persimmon (*Diospyros kaki* Thunb.) and banana (*Musa paradisiaca* L.) can easily be found along the forest edge, in abandoned farm sites (Tijuca Massif) and intermixed with the forest in small orchards and plantations (banana and persimmon in the Pedra Branca Massif; figure 6c–f) [51,52,58,74].

It is important to highlight that although species diversity is lower and composition is very distinct from its original state these novel ecosystems recovered forest biomass and functionality. In these novel ecosystem sites, basal area (an indicator for biomass), is as large or even larger than conserved or

---

[8]*Carioca* is the Portuguese term, of indigenous *Tupi* origin, meaning a person, or people, from the city of Rio de Janeiro. *Carioca* culture is the modern melting pot of different cultures and traditions (western, afro-Brazilian, native-Brazilian and even eastern) currently composing Rio de Janeiro.

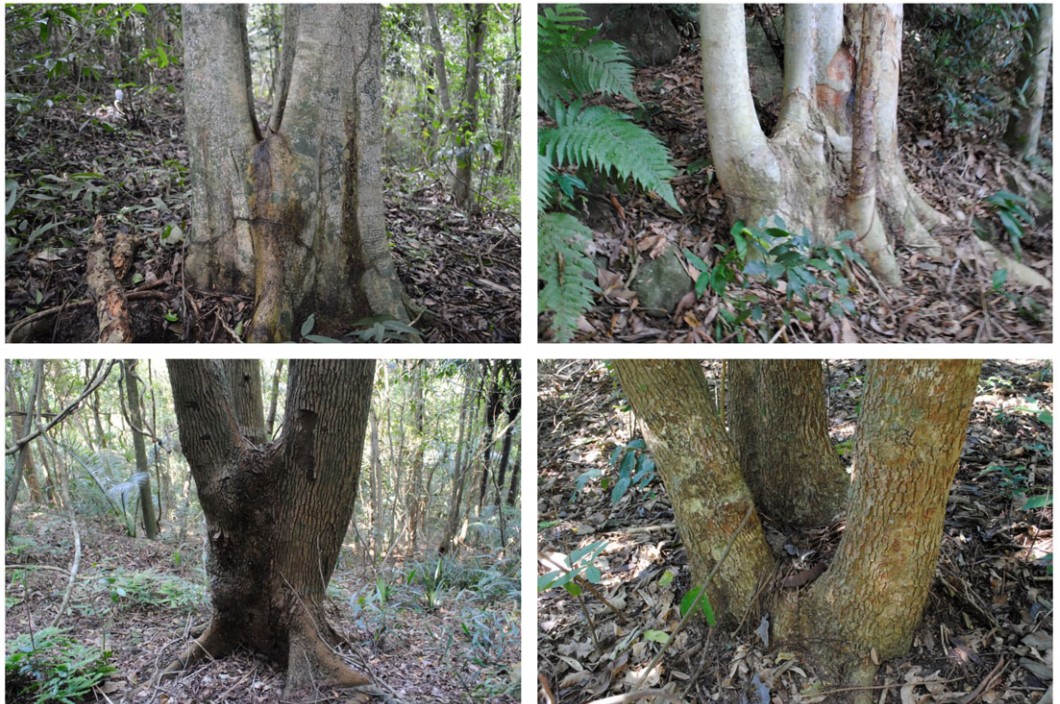

**Figure 5.** Resprouting of trees owing to previous felling, for charcoal production.

historical forest tracts, reaching up to 118 m$^2$ ha$^{-1}$. The participation of the jackfruit in the biomass resilience[9] of the forest shows how valuable this species can be to the system. Retaining and expanding biomass, sequestering carbon from the atmosphere, providing tree cover and forest stratification, in addition to providing a rich and abundant food for both fauna and humans, are some of the important ecosystem services provided by these novel ecosystems [17,58].

Vegetation also changes in subtle ways, not only from exotic species introduction, but with modification of native species distribution, density and dominance pattern. Ecological novelty includes never before seen patterns of species composition and relative abundance in a given biome [26], that comprises anomalous distribution of native species, which may present super dominant populations. In the Pedra Branca Massif, *Guarea guidona* L. (Sleumer), a native early-secondary tree species shows a pattern of high longevity, density and super dominance in sites of previous charcoal production [66,76,77]. The anthracological analysis that was made in an area where the species is currently dominant showed that the process of charcoal production that happened about 70 years ago caused a great reduction in the diversity of other species and increased density of *G. guidonia* [73]. Because of its longevity, reaching over 150 years, *G. guidonia* can be used as an indicator species for areas used for charcoal production and agriculture in the nineteenth century, when presenting elevated density and dominance.

## 3.4. Energy and nutrient flow

Another aspect of the forest exploration concerns the energy produced from the charcoal workers' labour. In global terms, its production reaches very significant numbers, considering only the 1522 charcoal plants discovered so far in both massifs. Assuming that each one has been used on average eight times [60], the total production of these charcoal workers was around 15 000 tons of coal. Its conversion into energy can be found in table 1. This massive production clearly shows the importance of the charcoal producer's work for the energy supply of Rio de Janeiro up until the mid-twentieth century when the country transitioned into a petrol-based energy matrix.

In the absence of studies referring to the study area we used data from nearby areas, obtained from the indicated research [78–80]. The order of magnitude of the nutrient return time is depicted in table 2.

---

[9]Biomass resilience is an ecological concept that explains how secondary tropical forests (i.e. recuperating from any type of anthropogenic disturbance) are able to recover biomass (tree density and tree size), returning to 90% of reference carbon stocks in 66 years, with more than 50% of biomass return in 20 years [75].

**Figure 6.** Illustrations of the novel-hybrid-managed ecosystems mosaic currently found at the Tijuca Massif (TM) and Pedra Branca Massif (PBM). (*a*) The ruins of an abandoned coffee farm from the nineteenth century (TM); (*b*) jackfruit novel ecosystem (TM); (*c*–*e*) current landscape with banana plantations intermixed with forest (PBM); (*f*) persimmon plantation embedded in the forest structure (PBM); and (*g*) dwelling of small farmer (PBM).

**Table 1.** Estimation of charcoal volume and energy content provided by the charcoal kilns studied in the Pedra Branca and Tijuca Massifs.

| total production (tons) | charcoal volume (m$^3$) | energy content (kcal) |
| --- | --- | --- |
| 15 426 | 77 130 | 113 609 425 200 |

The recovery of nutrients exported via charcoal production takes place in a time period between 10.5 to 15.9 years. In other words: theoretically, the functional recovery of the ecosystem takes place in less than 16 years. This estimate can simplify complex processes and local realities, which can alter these times. However, it brings evidence that the forest system recovers in functional terms after the removal of firewood for charcoal production.

## 3.5. Social-ecological system analysis

In order to better understand the complex social and ecological interactions and legacies that involved charcoal production in the forests of the Tijuca and Pedra Branca massifs, we applied the social-ecological systems framework proposed by McGinnis & Ostrom [81]. First, it is important to highlight

**Table 2.** Return time of nutrients (nitrogen, phosphorus and potassium) exported by the production of charcoal in the Pedra Branca Massif.

| export of nutrients (a) (kg ha$^{-1}$) | N | P | K |
|---|---|---|---|
| | 886.5 | 157.7 | 1304.8 |
| nutrient inputs (b) | N | P | K |
| atmospheric inputs (kg ha$^{-1}$ yr$^{-1}$) | 6.6 | 11.4 | 25.1 |
| litter inputs (kg ha$^{-1}$ yr$^{-1}$) | 78.1 | 0.8 | 57.0 |
| total (kg ha$^{-1}$ yr$^{-1}$) | 84.7 | 12.2 | 82.1 |
| cycling time (a/b) (yr) | N | P | K |
| | 10.5 | 12.9 | 15.9 |

that this application has some historical gaps since the charcoal production was carried out by a highly invisible population to the Brazilian society, made up of formerly enslaved populations of African descent with no formal historical records of their activity [59]. Second, the social-ecological systems framework was not designed to be applied to past social-ecological interactions and dynamics. However, we were able to work through these points and the application of the framework was able to not only shed light on nineteenth century issues but also helped to contextualize the land use legacies of this past event and understand current social-ecological dynamics related to novel ecosystems, biocultural diversity and small forest-dwelling farmers.

We synthesized the first-tier variables (resource systems, resource units, governance systems and actors) and subsequent second-tier variables with concise descriptions in table 3. Our analysis revolves around the charcoal workers (actors) felling small and medium sized trees (resource unit), in a governance system of conflicting interests, with enforcement of forest protection legislation at the same time that the demand for charcoal increased in the nineteenth century. This made the activity barely legal and clandestine, carried out by marginalized and socially invisible workers. This aspect alone is a strong indicator of the lack of social visibility and empowerment of charcoal workers in the context of nineteenth century *carioca* society. We elaborated a schematic representation of this past social-ecological system (figure 7) in order to visually simplify how forest biomass recovered from charcoal production to compose the complex forest-city social-ecological system of today.

As seen, charcoal production in the forests of Rio de Janeiro presented two aspects regarding its understanding as a social-ecological system. On one hand, it allowed the forest to return at least in terms of biomass and functionality. In terms of composition, there has obviously been a loss of numerous species. The forest as a whole is in ecological succession, in different stages and with different degrees of compositional novelty. Tree species diversity is greater in areas where the forest had more time to regrow after charcoal manufacture took place [62,82]. In sites that charcoal production occurred in the twentieth century, where the exploration took place 80 years ago or less, there is less diversity. The diversity of the forest at the time it was felled (basically in the late nineteenth and early twentieth centuries), was richer with a distinct set of species than the current regenerating forest. Patzlaff [67] carried out an anthracological research (study of historic charcoal) and identified tree species that were burned in the past, from charcoal fragments found at historic charcoal production sites. Comparing this list with the current tree species composition, the study concluded that the taxonomic diversity suffered a reduction of 40–60%. In other words, there was a return of tree biomass that was not accompanied by the recovery of forest diversity. Although the forest has recovered owing to the process of ecological succession that occurred after the abandonment of charcoal activities, the ecosystem has kept marks of this history in many of its attributes, such as long living early-secondary native and exotic species (*G. guidonia*, and jackfruit, for example). In summary, from a strictly biological point of view, the forest recovered after the abandonment of charcoal manufacturing, with biomass resilience and the maintenance of structure and functionality but modification of species composition (figure 7).

Regarding economical aspects, this activity was essential for urban-industrial growth, considering that the massive amount of charcoal produced was able to feed the expanding metropolis of Rio de Janeiro, especially at the turn of the nineteenth to the twentieth century. In this sense, the social-ecological metabolism of the city was completely interconnected with the forest dynamics and

**Table 3.** Charcoal producers – Forest Social-Ecological System in the nineteenth-twentieth centuries. (First-tier (S, RS, RU, A, GS, I and O) and Second-tier variables (RS1-9, RU1-7, A1-9, GS1-4, I1-9 and O1-3). The Social, Economic and Political Setting (S) are highlighted in grey and at the top because they are overarching for all other variables. Interactions (I) and Outcomes (O) are highlighted in grey and at the bottom because they are a result of the relations (setting conditions; being part; defining rules) between Resource System (RS), Resource Unit (RU), Governance System (GS) and Actors (A). Source: adapted from McGinnis & Ostrom [81].)

| Settings (S) |
|---|
| (S1) A growing capitalist economy based on commodity exportation and slave labour; (S2) High population growth rate, income concentration, high social and racial inequality; (S3) Relative political stability due to a central authoritarian regime; (S4) Increasing policy regulation of forest resource use due to depletion of water resources; (S5) High demand for charcoal from industry, locomotives, residential use and urban construction; (S6) Social invisibility of charcoal workers |

**Resource System (RS):** Forest and water resources of Rio de Janeiro remnant forests

**RS1 - Sector:** Timber and charcoal

**RS2 - Clarity of system boundaries:** Geographic limits of Pedra Branca and Tijuca massifs

**RS3 - size of resource system:** large - Pedra Branca 16.900 ha; Tijuca 11.500 ha

**RS4 - Human-constructed facilities:** Charcoal production sites; Farm and house ruins; Road network; Water catchment system

**RS5 - Productivity of system:** The forest presents an aboveground biomass range of 194.4–348.7 $m^3$/ha

**RS6 - Equilibrium of properties:** Forest thinning allowing forest regeneration and tree-stump sprouting

**RS7 - Predictability of Systems dynamics:** Forest structure regeneration after 20–30 years

**RS8 - Storage characteristics:** timber and useful plants storage (biomass)

**RS9 - Location:** the slopes above 50 m of the forest massifs

**Resource Units (RU):** Trees (small and medium size);

**RU1 - Resource mobility:** Trees are not mobile; however, charcoal is a highly compact and mobile resource;

**RU2 - Growth or replacement rate:** 25–30 years of forest biomass recovery (including tree-stump sprouting)

**RU3 - Interaction among resource units:** Climatic and hydrological interactions

**RU4 - Economic value:** Timber and charcoal were the energy matrix of the city until 1950. It was cheap to produce and had a low cost for consumers.

**RU5 - Number of units:** 28.500 ha of forest led to an estimated 4600 charcoal production sites found in the massifs.

**RU6 - Distinctive markings:** Charcoal kilns were made on plateaus on the forest slopes, leaving a distinct feature in the landscape, currently visible

**RU7 - Spatial and temporal distribution:** The charcoal production sites are widespread in the forest massifs

**Actors (A):** Charcoal workers

**A1 - Number of actors:** Unknown amount

**A2 - socioeconomic attributes of users:** Ex-enslaved and descendants; Poor, discriminated and socially invisible

**A3 - history of use:** Sugar cane Mills (17th–18th centuries); Coffee plantations (19th centuries); Small farms (19th–20th centuries);

**A4 - Location:** Spread out in small dwellings in the forest; Living in clandestine *quilombola* (ex-enslaved) communities in the forest

**A6 - Norms/social capital:** Cultural values related to the forest; sacred species (*Ficus* sp.) spared from cutting; territorialization and toponymy

**A7 - Knowledge of social-ecological systems/mental models:** Awareness of forest structure and ecological processes (felling only trees up to a specific size, sparing large trees); charcoal production *savoir faire*

**A8 - Dependence of resource:** Charcoal workers highly depend on forest resource for their livelihoods; Also depend on hunting and food production in the forest

**A9 - Technology used:** Broad axe and fire

**Table 3.** (*Continued.*)

| Settings (S) |
| --- |
| (S1) A growing capitalist economy based on commodity exportation and slave labour; (S2) High population growth rate, income concentration, high social and racial inequality; (S3) Relative political stability due to a central authoritarian regime; (S4) Increasing policy regulation of forest resource use due to depletion of water resources; (S5) High demand for charcoal from industry, locomotives, residential use and urban construction; (S6) Social invisibility of charcoal workers |

**Governance Systems (GS):** Legislation controlling forest use in order to diminish water scarcity (19th century); growing demand for charcoal as an energy source.

**GS1 - Government organizations:** Lack of government control and regulation of charcoal production

**GS3 - Network structure:** Charcoal outflow and transportation to the city centre intermediated by middlemen;

**GS4 - Property-rights systems:** Lack of property legislation for ex-enslaved populations

**Interactions (I)**

**I1 Harvesting levels of diverse users:** An average of 26,13 $m^3$ of timber used per charcoal kiln

**I2 Actor mobility:** Actor mobility is necessary in order to explore different locations for their timber

**I3 Knowhow sharing:** Knowledge by oral tradition and through social interaction

**I7 Self-organizing activities:** Charcoal producers organized in small 3–4 person groups

**I8 Networking activities:** Charcoal outflow and transportation to the city

**I9 City–Forest energy metabolism interaction:** In the city energy and population are concentrated; in the forest energy and population are scattered; Solar energy stored in forest biomass was transformed in a concentrated energy source (charcoal) and made available for urban population use

**Outcomes (O)**

**O1 Social performance measures:** Economic exploitation of marginalized populations; Lack of social visibility and empowerment: descendants live in the city slums in poor conditions

**O2 Ecological performance measures:** Forest recovery from the charcoal manufacturing activity has been verified over time (for structure but not for diversity/composition)

**O3 Externalities to other social-ecological systems:** Some descendants become small subsistence farmers producing banana and persimmon in the Pedra Branca Massif; *Favelas* (poor slum communities) occupy the fringes of both massifs with varying degrees of social-ecological interactions of residents with the forest (e.g. jack fruit extraction, religious use of forest and recreational activities); Currently both massifs became protected areas, with their own set of rules, regulations, inputs and outputs

charcoal producers. However, with regard to the social conditions, practically nothing happened to the main actors involved in the entire process (the charcoal producers). They did not receive the social benefits of a growing city; on the contrary, they were cast aside as a population living in the fringes of a booming society. Most of the charcoal workers' descendants currently live in the city slums and are linked to the poorest sections of the population, with a low Human Development Index. Some still live in small rural communities or as isolated forest dwellers (figure 6g). It is necessary to understand the social-ecological system as a whole; that is, understanding social relations and also the biotic and abiotic world. The transformation of these forested mountain landscapes into natural parks (Tijuca National Park and Pedra Branca State Park) helped to crystallize, simultaneously, the resilience of the ecological system and the lack of memory in relation to the work incorporated in it. Of that history, the only document that remained are the landscape marks etched by the hands of poor charcoal workers, today transformed into beautiful natural parks, which hides, in the middle of a dense forest, an intense history of invisibility and inequality of an important segment of Rio's society.

Today this complex social-ecological system of Rio de Janeiro can be understood as a result of hundreds of years of humans interacting with the forest. These urban forests are a complex mosaic of novel and hybrid ecosystems, together with managed lands and well conserved forest tracts, with people living in close contact to the forest and even within the large forest remnants. In other words, they are cultural forests with a rich history hidden beneath their canopy.

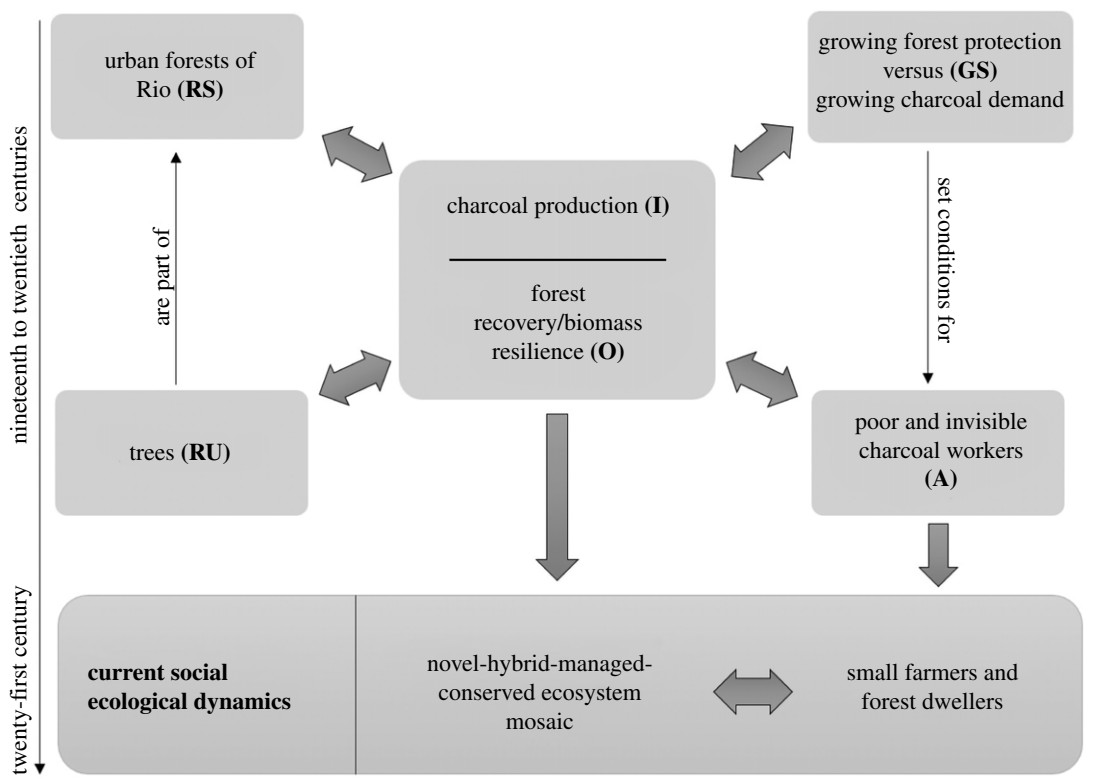

**Figure 7.** Schematic representation, based on McGinnis & Ostrom [81] revised social-ecological systems framework with multiple first-tier components—resource system (RS), resource unit (RU), governance system (GS), actors (A), interactions (I) and outcomes (O)—, of the Charcoal-Forest social-ecological system in the nineteenth to twentieth centuries, in the urban forests of Rio de Janeiro, with the current social-ecological dynamics in the twenty-first century.

The main takeaways from the social-ecological system analysis (organized in table 3 and figure 7) are: (i) the forest landscapes are rich historical-ecological documents of former charcoal production; (ii) the charcoal mode of production (management techniques, intensity and spatial-temporal scale) allowed for forest recovery/biomass resilience though with changes in species composition; (iii) novel ecosystems can be related to charcoal production sites, and understanding this can help forest management; and (iv) the descendants of charcoal producers either live in *favelas* or *quilombos*, on the fringes of the forest massifs, or are currently forest dwelling subsistence farmers, all of whom are living in varying degrees of socio-economic vulnerability.

## 3.6. Current social-ecological dynamics

According to Oliveira & Fernandez [83], bananas and persimmons are the most important products for the livelihood of the farming families that still inhabit part of the Pedra Branca Massif (figure 6c–g). Both are considered market crops and are grown together with other products that guarantee self-consumption or can also be sold. Banana trees and other crops (such as persimmon and manioc), in addition to guaranteeing the farmers social reproduction, also fulfil the role of marking the existence of farmers in the forested landscape. These hybrid forest-orchard systems are formed by paths, wooden houses, fences and pack animals. An in depth reflection regarding the coexistence between natural and agricultural environments implies considering the complexity of agricultural systems with all their elements—plants and animals, objects, food, techniques, forms of knowledge and space— since *in situ* conservation of agrobiodiversity represents the protection of cultural identities, social territories, local knowledge systems and corresponding ways of life. Also from the perspective of biocultural diversity, the *savoir-faire* and the cultural wealth of agricultural management techniques are also in themselves worthy of protection and heritage. It is important to note the efforts of flexible (and perhaps not purposeful) management of the Pedra Branca State Park to incorporate *quilombo* communities and areas of persimmon and banana production, important biocultural forest products, within its limits, albeit with some territorial conflicts unsolved. Thus, the complex mosaic of novel-

hybrid-managed-conserved forest systems provides not only livelihood and sustenance for forest dwelling families but also important ecosystem services, such as areas for recreation and religious practice [84], soil protection and water, food for urban dwellers and for local fauna.

In the Tijuca Massif, we are met with other landscape legacies and challenges. Considered by many ecologists and park managers as being an invasive species [85], jackfruit is distributed in many areas of the Tijuca Park: along highly frequented trails and roads, along former charcoal production sites and especially in the edges of the forest with urban areas, many times in the 'backyard' of *favela* communities. The understanding of the historical and biocultural aspects of current jackfruit distribution in the forest, which developed into novel ecosystems, can be helpful for Tijuca National Park managers. Because jackfruit novel ecosystems have surpassed a social-ecological threshold and are a part of the forest ecosystem (not being possible to eradicate its presence from the Tijuca National Park), management has focused on tree-barking techniques (i.e. killing adult trees) to control jackfruit populations from expanding towards core areas of the park. However, another management technique for population control would be to remove the fruits from jackfruit trees in critical areas, impeding new jackfruit seedlings and saplings from establishing and growing.[10] A second step of this novel ecosystem management proposal would be to benefit local communities with the collected jackfruits, and even carefully planning and regulating jackfruit removal for personal consumption in communities that border the forest. This could address ecological goals (controlling the expansion of an exotic species for conservation management) and social objectives such as food security and community empowerment, which are also important goals of the biocultural diversity perspective. By applying a non-dichotomous framework—such as social-ecological systems, novel ecosystems and biocultural diversity—, there is a vast opportunity to control what is considered a harmful invasive species while benefiting local livelihoods, through an 'eat it to beat it' style of management [86]. This would help shift back *carioca* perception of jackfruit, to its earlier historical view of a useful tree that offers abundant fruits. As vegetarian diets, local sustainable food consumption and agroforestry food production are becoming more popular in Rio de Janeiro, jackfruit has ample space for a growing market of conscious consumers.

# 4. Conclusion

After almost two centuries of regeneration, the legacies of past human activities are observable in the structure, composition and functionality of the forest. The interaction of these land uses with natural processes and the intrinsic properties of the landscape combined to create the forest mosaic existing in the present. We can argue that, although not purposeful, the forest management carried out by the charcoal producers, in its spatial and temporal settings, facilitated ecological recovery (especially biomass, forest structure and functionality), yet socially did not improve their own livelihood and wellbeing, as they remained invisible and marginalized. The artisanal technique used (with a simple axe and fire), the low population density, the spatial diffusion of its practice, their ecological knowledge about the forest, the practice of sparing large trees (out of reverence, spirituality or owing to labour difficulty), as well as the natural regeneration coupled with the introduction of exotic species (for food, timber, medicinal and ritualistic purposes), leads to the conclusion that charcoal production by the marginalized population of the nineteenth and early twentieth centuries facilitated forest recovery and biomass resilience, at the cost of their own invisibility and marginalization.

Other practices that have occurred in forested areas, such as coffee production or animal husbandry, left disastrous land use legacies. In the first case, an effort to expropriate coffee farms and the implementation of the first reforestation project in Brazil was necessary to induce and assist forest regeneration. In the second case, the social-ecological legacies led to a total alteration of the structure, composition and functionality of the system that remained 'frozen' in an alternative state of plant succession, where exotic grasses dominate the landscape. Few trees are able to survive in an environment that has become prone to periodic fires, resembling it to a novel savannah-grassland

---

[10]Recently a very successful collaboration between a local jackfruit social-environmental enterprise (Mão na Jaca), civil society, association of cyclists and Tijuca National Park management, removed potentially dangerous overhanging jackfruit from trees covering a popular cycling road to one of the most popular viewpoints (Chinese View). After the two main goals were accomplished (jackfruit population control and protection of cyclists from potentially death threatening falling jackfruits) the collected fruits were distributed to nearby *favelas*, thus beneficiating the communities with a rich source of food. This novel management action of jackfruit removal occurred after a cyclist was hit on the head from a falling jackfruit on the road to the Chinese View. The details were personally communicated to us by the parties involved.

ecosystem, notably in the northern slopes of the two forest massifs. Reforestation efforts in the north facing slopes of Tijuca and Pedra Branca massifs have a low success rate and is further hindered by the difficult access to these areas, which are surrounded by drug traffic controlled *favelas*.

Charcoal production was the main source of energy and played a central role in the construction of Rio de Janeiro's social-ecological history. This chapter of history is slowly coming into light through recent publication of historical geography and historical ecology. These interdisciplinary disciplines, which are now part of the emerging field of environmental humanities, have raised important awareness on how forest and society are interconnected and interdependent, in the past and present, providing useful information for policy makers and park managers. The application of the social-ecological systems framework helped to elucidate some points of interconnection between the charcoal workers and the forest, not only as energy suppliers to the city, but as important producers of their living and working domains, generating palaeo-territories and novel ecosystems that today we find hidden in the forest. Although this paper considers the practice of charcoal production in the nineteenth to twentieth centuries to allow for forest natural regeneration, we do not advocate that this practice be resumed today. On the contrary, the forest reserves that were implemented as a national and state park have an important role in maintaining important ecosystem services for the city, conserving the rich remaining biodiversity, sheltering native (and exotic) species of flora and fauna and spaces for recreation and spiritual practices with the forest.

Exotic species are often seen as inherently bad or negative to the ecological system, incorporating conservation values more often than scientific evidence. For instance, jackfruit is at the centre of this debate, where many environmentalists consider it to be an invasive species that needs to be eliminated from the forest in order to achieve the historical baseline of species composition (before the arrival of Portuguese colonizers). However, the overwhelming evidence confirming that humans are changing Earth in unprecedented ways (i.e. the Anthropocene) has led to the understanding of shifting historical baselines (of biotic communities and abiotic environments). Therefore, the decision for any specific baseline is a political one, based on a specific set of values (such as pristine nature) and not a scientific one [87]. A social-ecological system or a biocultural approach can view this 'problem' as an opportunity to understand the cultural and historical legacies of past populations living in the forest and using its resources, and that the high density and dominance of exotic species found in the novel ecosystem forest tracts have a lot of hidden history beneath their canopy. In terms of sustainability, it would be far more advantageous to harvest the jackfruit from these novel ecosystems, controlling its expansion in a protected area, generating income and dealing with issues of food insecurity in the slums of Rio, instead of an approach that tries to eliminate this fruit species from the forest, a practice that historically has been unsuccessful in Rio.

Novel and hybrid ecosystems are fundamentally social and ecological. They emerge from the intersection of self-regulated nature and human intervention. In addition to being partly social-cultural creations novel ecosystems provide many important goods and services, including places to connect with nature, sources of drinking water, resources for local livelihoods and refuges for different species [88]. According to Hallet *et al.* [27], there may be instances in which novel ecosystems are preferable to any historical ecosystem, because novel species assemblages are more resilient and are better able to respond to ongoing environmental and anthropogenic changes. The novel ecosystem paradigm can displace management concerns of the single objective of conserving pristine ecosystems towards a greater qualitative consideration of how ecosystems have also been co-constructed by past populations, are protected by their inhabitants [74] and work to provide the habitat of species and ecosystem services [27]. This paradigm can provide an important framework for managers who must set goals for restoration and intervention in an increasingly transforming world. Finally, the concept of novel ecosystems could redesign the way we think and practice conservation, remodel our interactions with nature and renew public dialogue on ecosystem management. This is in close dialogue with the biocultural diversity approach that shifts from a dichotomized and utilitarian conception of nature towards a systemic perspective inclusive of other worldviews and human–nature interactions [35].

We believe that these concepts and frameworks can offer practical solutions for urban forest management. Exotic fruit species are an inherent element of the biocultural diversity of Rio de Janeiro [17,89] that can provide an important source of food for the population (both marginalized and affluent) while also increasing awareness of sustainability and food security in socially complex cities and increase the possibilities for biodiversity conservation.

Data accessibility. The data used in the paper is secondary data from the authors previous publications and can be accessed in [52,57–62].

Authors' contributions. A.S. contributed substantially to the conception and design, acquisition of data, analysis and interpretation, drafting the article and revising it critically and giving final approval of the version to be published. R.R.d.O. contributed substantially to the conception and design, acquisition of data, analysis and interpretation, drafting the article and revising it critically and giving final approval of the version to be published. A.B.-M. contributed substantially to the conception and design, analysis and interpretation, drafting the article and revising it critically and giving final approval of the version to be published. All authors are in agreement to be accountable for all aspects of the work in ensuring that questions related to the accuracy or integrity of any part of the work are appropriately investigated and resolved.

Competing interests. We declare we have no competing interests.

Funding. This study was financed in part by the Coordenação de Aperfeiçoamento de Pessoal de Nível Superior - Brasil (CAPES) - Finance Code 001. R.R.d.O. is a productivity fellow of the CNPq (Brazilian Research Council).

Acknowledgements. We would like to thank all of our students and collaborators of the Biogeography and Historical Ecology Laboratory at the Geography and Environment Department. We also would like to thank all of the valuable comments and contributions from the reviewers that added important insights and enriched our paper.

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
