## [Peer Review File · Royal Society Open Science]

Review History

RSOS-201855.R0 (Original submission)

Review form: Reviewer 1

Is the manuscript scientifically sound in its present form?

Yes

Are the interpretations and conclusions justified by the results?

Yes

Is the language acceptable?

Yes

Do you have any ethical concerns with this paper?

No

Have you any concerns about statistical analyses in this paper?

No

Recommendation?

Accept with minor revision (please list in comments)

Comments to the Author(s)

I think this a very informative paper, which offers a synthesis of knowledge related to the historical effects of charcoal production on the forests within the municipality of Rio de Janeiro.

I have two main comments:

- 1) Although very well-written, I found the paper unnecessarily long. The Introduction has 4 paragraphs explaining similarities and differences in the approaches of distinct disciplines, which I think is hardly necessary for the readers of this journal.
- 2) I have concerns with the optimism of the authors with the 'novel ecosystem' concept. They have soundly defined it and I tend to agree with their arguments. However, the literature is filled with criticism to the concept that I think they should at least acknowledge (e.g., Murcia et al (2014) Trends Ecol Evol). To support their arguments, the paper by Evers et al. (2018) Glob Ecol Conserv can be useful.

One very minor issue is that socioecological is written in three different ways along the text.

Review form: Reviewer 2

Is the manuscript scientifically sound in its present form?

Yes

Are the interpretations and conclusions justified by the results?

Yes

Is the language acceptable?

Yes

Do you have any ethical concerns with this paper?

No

Have you any concerns about statistical analyses in this paper?

No

Recommendation?

Major revision is needed (please make suggestions in comments)

Comments to the Author(s)

My comments and suggestions are found in the attached file (Appendix A).

Decision letter (RSOS-201855.R0)

Dear Dr Solórzano

The Editors assigned to your paper RSOS-201855 "Land Use and Social-Ecological Legacies of Rio de Janeiro's Atlantic Urban Forests: from charcoal production to novel ecosystems" have now received comments from reviewers and would like you to revise the paper in accordance with the reviewer comments and any comments from the Editors. Please note this decision does not guarantee eventual acceptance.

Please submit your revised manuscript and required files (see below) no later than 21 days from today's (ie 06-Jan-2021) date. Note: the ScholarOne system will 'lock' if submission of the revision is attempted 21 or more days after the deadline. If you do not think you will be able to meet this deadline please contact the editorial office immediately.

on behalf of Dr Agnieszka Latawiec (Associate Editor) and Pete Smith (Subject Editor)
openscience@royalsociety.org

Associate Editor Comments to Author (Dr Agnieszka Latawiec):

Associate Editor: 1

Comments to the Author:

Please carefully incorporate all suggestions made by the reviewers especially those attached in a separate document by Reviewer 2.

Reviewer comments to Author:

Reviewer: 1

Comments to the Author(s)

I think this a very informative paper, which offers a synthesis of knowledge related to the historical effects of charcoal production on the forests within the municipality of Rio de Janeiro.

I have two main comments:

- 1) Although very well-written, I found the paper unnecessarily long. The Introduction has 4 paragraphs explaining similarities and differences in the approaches of distinct disciplines, which I think is hardly necessary for the readers of this journal.
- 2) I have concerns with the optimism of the authors with the 'novel ecosystem' concept. They have soundly defined it and I tend to agree with their arguments. However, the literature is filled with criticism to the concept that I think they should at least acknowledge (e.g., Murcia et al (2014) Trends Ecol Evol). To support their arguments, the paper by Evers et al. (2018) Glob Ecol Conserv can be useful.

One very minor issue is that socioecological is written in three different ways along the text.

Reviewer: 2

Comments to the Author(s)

My comments and suggestions are found in the attached file.

===PREPARING YOUR MANUSCRIPT===

===PREPARING YOUR REVISION IN SCHOLARONE===

Author's Response to Decision Letter for (RSOS-201855.R0)

See Appendix B.

Decision letter (RSOS-201855.R1)

Dear Dr Solórzano,

It is a pleasure to accept your manuscript entitled "Land Use and Social-Ecological Legacies of Rio de Janeiro's Atlantic Urban Forests: from charcoal production to novel ecosystems" in its current form for publication in Royal Society Open Science. The comments of the reviewer(s) who reviewed your manuscript are included at the foot of this letter.

You can expect to receive a proof of your article in the near future. Please contact the editorial office (openscience@royalsociety.org) and the production office (openscience_proofs@royalsociety.org) to let us know if you are likely to be away from e-mail contact – if you are going to be away, please nominate a co-author (if available) to manage the proofing process, and ensure they are copied into your email to the journal.

on behalf of Dr Agnieszka Latawiec (Associate Editor) and Pete Smith (Subject Editor)
openscience@royalsociety.org

Appendix A

Summary

This is a timely, interesting, and well-written paper based on a rich empirical material.

It looks at a marginalized and little-documented historical population (charcoal workers) and trace their influence in past and present social-ecological interactions in two large urban forest remnants (now protected areas) in Rio de Janeiro. While nature conservation policies have often departed from ideas of “pristine nature” separated from humans, the study highlights how previous small-scale charcoal production helped shape the now-protected areas into “novel and hybrid ecosystems” with recovered biomass and functionality (albeit less species diversity – distinct from original state). In doing this, the authors outline the scientific “detective work” to uncover historical livelihood practices through the marks they left on the landscape.

The paper, which makes a decade of research accessible to non-Portuguese speaking audiences, has the potential to feed into various scientific debates around conservation, human-nature relations, ecological practices of traditional/marginalized populations, and biocultural diversity in urban areas. The use of the Social-Ecological Systems framework seems suitable (although some concepts are not defined, see below). However, some more work could be done to make the paper live up to this potential, mainly regarding problem framing/novelty (Introduction) and presentation (Sections 3.5-3.6).

Below I first outline three major concerns, and then list more detailed/superficial comments in the order of appearance in the draft. The draft could also use a language check (I have highlighted some issues that caught my attention – small things that together affect the rating of the paper – but I might have missed others). If the authors could address the concerns cited, I would warmly recommend the publication of the paper.

Major comments

I have three overarching (and interlinked) concerns: 1) a more clearly articulated research question (& associated research gap) would better motivate how the paper contributes to advancing science, 2) some terms could be better defined/explained for an international/interdisciplinary audience and 3) I am not fully convinced by how the paper employs the term sustainability.

1) Novelty? Research gap and research question can be more clearly articulated

This is my biggest issue with the paper’s presentation. Due to the focus and method (historical processes & mostly secondary data) it would be important to clarify what is the novel element: Which present-day gap/problem in scientific knowledge does it aim to address, and why is it important to (re)publish with an SES lens? (Part of the answer is found between the lines: The marginalized charcoal workers were little documented and bibliographic references of historical charcoal production in Latin America are still scarce, previous were published papers in Portuguese... but I suspect there is more.)

The introduction starts by describing developments in different fields (SES, sustainability/social research, environmental history etc.), which are presented as “tools” for the research, however, it

does not clearly connect this to a lack or a gap in knowledge in any of the areas, to make way for a research question.

Also, only a very general aim is presented: “to analyse the landscape transformations that occurred in the urban forests of Rio de Janeiro in the last 200 years in order to comprehend current social-ecological interactions and outcomes from this historical-ecological process”. (What scientific or practical problem could “comprehending social-ecological interactions” help to solve?)

I believe that formulating 1) a more specific (theoretical) research gap that the paper aims to fill, and consequently, 2) a more specific aim and research question, would help with both the aspect of novelty, and that of structure (i.e., with one or more specific questions formulated in the intro, it would be easier to understand how the results section is structured to answer the question(s).

There are plenty of interesting material in the result section, so the easiest way would perhaps be to work “backwards” from there and craft research question(s) at the end of intro that give credits to the findings. Possibly one methods-oriented question that valorizes the empirical “detective work” to get at these historical practices (current sections 3.1-3.4). And one regarding the application of the SES framework (3.5-3.6), e.g. something like:

1. What does (geographical analysis of) the landscape tell us about the historical/ecological (?) practices of the coal workers?
2. Some question that relates to their role in (transformation towards) the present-day novel ecosystem/social-ecological system (that requires the SES analysis and addresses a gap in research)?

Based on this, the aim could be made more specific, possibly mentioning sth like “Tracing the role of socially invisible charcoal workers” (The excellent paragraph on p. 10, row 18-25 might inspire the language for general framing of the paper.)

From there, the authors could go another step backwards to identify the research gap that the questions fill (i.e., which literature/problem do they want to feed back into?). Some ideas:

- The following two sentences could be developed into a paragraph with some more details and references: “Understanding how societies used natural resources in the past, producing landscapes imprinted with our history, has become a very important field of scientific inquiry in recent decades. Without having a good knowledge about how the landscape transformation process took place, one cannot make good management decisions regarding its current use, resource management and biodiversity conservation.” (E.g., what is considered “good knowledge” or “good management”, why do we have poor knowledge or management, what are practical or scientific gaps or hurdles (including recent references that say this is a gap) or what could this knowledge achieve on the ground for sustainability, land use, or marginalized populations...?), AND/OR:
- Is there any problem/drama/disagreement...? regarding NOVEL ecosystems and conservation?, AND/OR:
- Do the authors want to say that their approach drawing on Historical Ecology, Environmental History and Historical Geography to make a historically grounded SES analysis is the novelty, then they should perhaps write 2-3 sentences here to give a brief overview of other historical SES studies and what they did or didn’t achieve. (E.g., while aspect x and y have been considerably studied, no one looked at aspect z.), AND/OR:

- Biocultural diversity is briefly mentioned in section 3.6. IN CASE authors want the results to speak to ongoing debates on biocultural diversity/biocultural landscapes, here are some recent studies that might formulate related research gaps: Hanspach, J., et al (2020). Biocultural approaches to sustainability: A systematic review of the scientific literature. *People and Nature*, 2(3), 643–659. <https://doi.org/10.1002/pan3.10120>. Velázquez-Rosas, N., E. et al. 2018. Traditional Ecological Knowledge as a tool for biocultural landscape restoration in northern Veracruz, Mexico: a case study in El Tajín region. *Ecology and Society* 23(3):6. <https://doi.org/10.5751/ES-10294-230306>.

2) Terms and their definitions

Another (more easily addressed) issue with the manuscript is that the authors use some terms without explaining or defining them, including “resilience”, “biomass resilience”, “geographical tableau”, “ecosystem services” (not sure if this one is considered self-explanatory these days), “macro vestiges”, etc. Some of these might stem from the translation from Portuguese. To help the reader (and save them from having to google unfamiliar terms) and for the paper to reach its full potential of international and interdisciplinary spread, I would recommend defining (even if only with a few words) the technical terms necessary for the paper’s argument and using simpler words for those that are not. See detailed comments for examples.

3) Sustainability/unsustainability

My third critique regards the structure and focus of Sections 3.5-3.6: Why is it important to discuss (and emphasize as a key result in the abstract) whether the historical coal production practices were ecologically, socially, and economically SUSTAINABLE? It seems unfair to judge past practices against a modern sustainability lens – and it is not clear from the paper’s framing what it is supposed to achieve. (It gives the impression that Section 3.5-3.6 were written to answer a research question about sustainability, while such a question was never posed.)

This use of the term sustainability feels somewhat like a detour (and awakens questions such as the climate impact of the estimated 15,000 tons of coal produced – would it still be ecologically sustainable considering our current knowledge of anthropogenic climate change?). Is the sustainability “assessment” an explicit part of the SES framework used? Or would it not be enough to describe the practices’ importance for city/development AND positive effect on the forest? (While I’m not an expert on terms like metabolism or interconnectivity, it sounds more logical to me to phrase it in such terms, as in this sentence: “the social-ecological metabolism of the city was completely interconnected with the forest dynamics and charcoal producers”.)

Regarding the “social unsustainability” of the workers’ conditions and marginalized position, I also find this very relevant, although I am not convinced about the cause and effect: do the authors mean that they became MORE marginalized through the coal work? (or was it rather something they resorted to because there were few options, and which granted them some level of self-sustenance/autonomy). Rather than putting a label of sustainable/unsustainable, it might be interesting to know something more (if possible) about the work and the exchange, did their marginalized position and the clandestine nature of the work also mean that they had to sell the coal at suboptimal/ exploitative prices (etc)?

In relation to this, the description of the historical contribution to the city and to the forest (section 3.5) and the present-day traditional/forest/quilombola populations (Section 3.6) appear rather isolated, even though Fig 7 suggests there is a link. Could the authors possibly make the

link/significance of the historical findings to present-day people/biocultural diversity/conservation/land use more explicit? (From work in a related area I am wondering whether the findings could possibly speak to present-day titling processes of such populations under pressure from both conservation legislation and extraction/urbanization, but I'm not sure this is what the authors want to get at?)

A more specific and analytical research question, grounded in a research gap and covering the SES analysis and sections 3.5-3.6 (see overarching concern 1), might help to productively rearrange these sections to answer the question!

Detailed comments – aimed at further increasing clarity of the paper

ABSTRACT

1-2 introductory sentences would help to understand up front what research gap the paper wishes to fill, e.g. starting with: “Historical ecology has been an important tool in deciphering human-environment interactions imprinted on landscapes throughout time, BUT it has been little applied to.../ there is lack of knowledge on...”

The use of sustainability is a bit confusing (see overarching concern 3).

INTRODUCTION

Social-Ecological Systems/social-ecological system/Social-Ecological system/SES – should be consistent (I agree writing it out is the more elegant choice and easier for the reader).

“adaptive landscape” – possibly confusing since the term adaptive can have different meanings across ecology, SES and the more socially-oriented sustainability science. The same is true for terms like “individuals” and “communities” (I spotted the latter used in two different meanings in the article), which might call for specifications like “plant and animal communities” (or what is most suitable) to avoid confusion. AND resilience (would be important to know whether the authors are using it in line with ecological or social-ecological literature)!

“Social Ecological Systems (SES) has been the main theoretical and methodological framework used in the emerging field of Sustainability Science” – can be debated! I would at least say “ONE OF the most important” and support it with a reference – such as the recent Clark and Harley (see eg its Table 1). There are different versions of the framework and many of the more social science-oriented Sustainability scholars are critical to this “biosphere based” Sustainability Science (they would also strongly object to the ensuing claim that SES as an important part of “social research” today – most social research still does not even consider the environment!).

“good knowledge” – sounds simplistic, good for what?

The role of the few paragraphs on pages 3-4 that talk about the three history-related fields (Historical Ecology, Environmental History and Historical Geography) is not fully clear. The authors use a language of “highlighting contributions” of fields that have (similarly to SES) “contributed to the construction of an approach that seeks to overcome the separation between nature and society”. They could take a more active stance here (do you draw on all three in the paper? Say so – and how! Is there a lack of integration between these fields and SES science? Say so – backed up by recent references!) If they are merely “conceptual tools” the authors draw on, but do not help formulating the research gaps/questions, perhaps consider putting them in a separate “Conceptual framework”

section (where it also might be relevant to define terms like “resilience” and “biomass resilience” and how they are linked).

Vestiges – is this a specific geography term? Would it be possible to use a simpler term throughout (such as traces?) – or else perhaps provide a definition and reference.

METHODS AND MATERIALS

Quilombos/Quilombola – the term first appears in methods section without definition. Possibly lift the explanation here from the later footnote, since not many non-Brazilians know about them and it is a recurrent & interesting feature of the paper.

“This paper synthesizes past studies conducted by two of the authors, in the last decade” – could already here add the references mentioned in the author-supplied statements (34, 39, 40, 41, 42, 43, and 44) and perhaps mention that they were published in Portuguese, which helps motivate the publication of this paper (=to reach a new audience).

“historical, archaeological, vegetation and spatial data” – could be vegetational for consistent form

“iconography”? Could the authors use a few words to explain what this means as a research method.

check wording and logic of this extract – “intense” lack and “clear” focus sound a bit off, and it reads like the present study IS charcoal production (rather than IS ON/ABOUT): “In the present study – charcoal production in the forests of Southeast Brazil – the lack of documental information is intense. For the charcoal manufacturing process, generic references are found, with a clear focus on metallurgy from 16th to 19th century, such as the work of Landgraf et al (37), or from the beginning of the 20th century (38).”

“Concerning sampling techniques” – confusing, I find that the paragraph does not explain about sampling techniques. What is the role of two different study areas, do their different features allow for some kind of comparison? (“also” in the next sentence is confusing – it reads as if the field methods are another sampling technique, is this the intention?)

“kiln” – could this term possibly be explained at first mention – e.g. by oven/production site or what would be the proximate word – and/or by referring to the excellent Figure 1?

“first-level” “second-tier” – Between this explanation, the result section, and Figure 3, I am a bit confused by the notion first/second (level)/tier (choose one?!) variable. First/second in relation to what scale? Adding a sentence here, and/or redesigning Figure 3 to elucidate this, might be useful to explain/exemplify the levels. I also wonder why some, but not all, of these abbreviations (e.g. U and ECO) show up in Table 3 and Figure 7.

RESULTS

Sections 3.1 - 3.4

“There is a limited amount of written documentation about charcoal production prior to the 20th century THAT CAN BE FOUND.” – last words are redundant

“Inscribed in the current landscape as signs or traces of different NATURES” – I suppose the authors refer to different “types/kinds”? Perhaps avoid using the term nature in other meanings than “natural environment” to avoid confusing the reader (“such as” before the colon can also be dropped to avoid repetition).

“the charcoal activity demanded very few initial inputs (of tools) and guaranteed relative independence” – I understand there is little documentation, but do the authors have any idea or estimated guess as to who bought the coal (if people in the city, how it was transported from the remote areas?) and what was offered as payment (money/goods?). Would be relevant to know to understand how it contributed to their subsistence.

“geographical tableau” – not even with the help of Google I could find an explanation for this term (in English). Can it be explained/defined, even if only with a few words? Following the reference, I get to Gomes’ concept of “quadros geográficos” – would perhaps be interesting to lift this into the text for the international reader, sth like “Borrowing a concept from Brazilian Geographer Paulo da Costa Gomes, such evidence constitutes today a ‘geographical frame’ (quadro geográfico), meaning.../referring to [geography as a graphic way of structuring thought]. This frame allows us to think...”

(I prefer frame over tableau since it has the same double meaning of theoretical frame(work)/lens and painting, see also <https://journals.openedition.org/confins/21686>)

“the worker’s management planning” – do you mean workers’ (plural)?

“Our research is still under development”???? – confusing, will you add more results, or do you mean that the paper is the first of a number of planned publications?

“In the Pedra Branca forest, 104 ruins were found and 107 ruins found in the Tijuca Massif.” – confusing sentence structure, using the same format/order for both would help the reader

“In the Pedra Branca Massif, although the vast majority of charcoal sites and ruins are located below 300 m altitude, only 30% are above this elevation.” – delete “although”, I see no opposition between the first and second statement (could be joined with semicolon) – whole paragraph (about the elevation) could be revised for consistency and clarity.

“steep, elevated and distant from level terrain” – seems to be a noun missing after elevated?

“it was a parallel and marginal occupation for the farmers and land owners”. – I suppose you mean “TO/(in the eyes of) the farmers and land owners, it was a parallel and marginal occupation.” – “for” sounds like they were the ones carrying it out.

“In part, this is due to a fact that may have played a role” confusing sentence, either it is a fact or a possibility, can be put more simply as → “The regrowth of stumps may have played a role in the return of the forest: some species that were cut for charcoal production are able to regrow and, since they lose apical dominance, they created bifurcations in the regrowth process (Figure 5) (60, 65).

“constitutes a macro vestige of past use of the forest” – what is a macro vestige? Can you use a simpler term? Or just delete the clause to avoid repetition with next sentence: → “This pattern is relatively common in areas that were deforested. The (!) resprouted tree stumps compose one of the vegetation indicators of past forest use alongside native species with anomalous distributions and persistence and dominance of previously introduced exotic species, especially fruit trees and ritualistic herbs (60).”

“and that was also consumed by the charcoal workers” - do you mean “it”?

“Jackfruit is rich in carbohydrate and protein and of low cost, it was potentially transported to the areas of its work in the forest, germinating from the uneaten remains and discarded in the forest.” – Check this sentence for repetition (low cost) and logic (should it be “their work” and “remains discarded”?) – check the rest of the paragraph for grammar/punctuation (e.g. “ON the edge”)

Carioca – word needs explanation to non-Portuguese speakers

Sections 3.5 - 3.6

(The first paragraph in 3.5 sounds a bit more like methods language to me.)

It could be clearer in this section what are the main takeaways from the impressive Table 3. (That address research gaps/contribute to new knowledge).

“biomass resilience”? term was never defined – does it relate to the SES framework or is it an ecological term?

“Ecological novelty includes never before seen patterns of species composition and relative abundance in a given biome (22), that include anomalous distribution of native species, which may present super dominant populations.” – repetition of “include”

“energetic sustainability” – do you mean energy supply? Why the need to call it sustainability

“with increase forest protection legislation at the same time that the demand for charcoal increased” – do you mean increased? (possibly use other word to avoid repetition, “as FPL was rolled out/enforced/... at the same time that”)

“Patzlaff (50) ... identified tree species that were burned in the past” – I would be curious of how many he identified and perhaps which were most common?

“The ecosystem has kept marks of this history in many of its attributes, SELECTING long living early-secondary native and exotic species” – do you mean “including”/“such as”?

CONCLUSION

“lighter” – is this the right word, would torch be better?

“at the cost of their own invisibility and marginalization” – does this mean they become MORE marginalized???? Not sure this is currently backed up.

“Reforestation efforts in the north facing slopes ... is further hindered” – should be “are”.

“In terms of sustainability it would be far more advantageous to harvest the jackfruit from these novel ecosystems, generating income and dealing with issues of food insecurity in the slums of Rio” – I agree (with the point and the use of sustainability)!!! this would be an interesting research project! (also corresponding with the ‘fitness’/organic/vegetarian trend we describe in our study on biocultural diversity in Rocinha) – although there is a conflict with a latter sentence mentioning jackfruit as an “important source of food for the population” – this makes it seem like this is already the case?

Argumentation in this last (long!) sentence is somewhat circular: “Although some management issues remain to be resolved, such as the management of exotic fruit species that still provides an important source of food for the population (both marginalized and affluent), it is noteworthy the efforts of flexible (and perhaps not purposeful) management of the Pedra Branca state park to incorporate quilombo areas and areas of persimmon and banana production, important forest products, within its limits, albeit with some territorial conflicts unsolved.” Is it really a concluding remark, it sounds more like new topics are being introduced (which might be more relevant to bring up in sections 3.5-3.6)?

TABLES AND FIGURES

The tables and figures are great and really help to illustrate the text!

Table 3 is impressive and gives an overview of the complex interactions! It is however not so intuitive from the table why some variables appear horizontally and others vertically (i.e. 1 tier, 2 tier). Perhaps including such information in the titles (or possibly using another layout) could help to see this? (Also check for consistent use of small/capital letters and hyphen/n-dash.)

If short of space for figures, Fig 3-4 could be combined into one (might also allow for better comparison).

Figure 7: Great figure! Would it be possible to repeat the abbreviations in the caption?

Appendix B

Manuscript: RSOS-201855 "Land Use and Social-Ecological Legacies of Rio de Janeiro's Atlantic Urban Forests: from charcoal production to novel ecosystems"

Authors: Alexandro Solórzano, Ana Brasil-Machado, Rogério Ribeiro de Oliveira

Special collection: Sustainable Land Use: successful initiatives and state of art

Response document for Reviewers 1 and 2

All indications of corrections made on the manuscript with page numbers and line numbers (**p. x line y**), are related to the document with Tracked Changes. This is to help the editors and reviewers to see all corrections that were made.

Response to Reviewer 1

Point 1: Although very well-written, I found the paper unnecessarily long. The Introduction has 4 paragraphs explaining similarities and differences in the approaches of distinct disciplines, which I think is hardly necessary for the readers of this journal.

Response 1: We thank Reviewer 1 for his/her comments. Since the paper synthesizes more than 10 years of research, it would be hard to write a shorter piece. We hope that our corrections and reviews have diminished its length. Regarding the introduction, we believe that the sub-divisions inserted in the first section (1.1. Social-ecological systems; 1.2. Historical Coupled Human-Environment Investigation; 1.3. Novel Ecosystems and Biocultural Diversity; 1.4. Human transformation of the Atlantic Forest) will help navigate the reader through the different disciplines and conceptual tools used in the paper. Since the paper's proposal is to connect historically informed environmental investigation (Historical Geography, Historical Ecology and Environmental History) with current cross-disciplinary conceptual frameworks (social-ecological systems, novel ecosystems, biocultural diversity), we find it necessary for the readers and should remain in the paper.

Point 2: I have concerns with the optimism of the authors with the 'novel ecosystem' concept. They have soundly defined it and I tend to agree with their arguments. However, the literature is filled with criticism to the concept that I think they should at least acknowledge (e.g., Murcia et al (2014) Trends Ecol Evol). To support their arguments, the paper by Evers et al. (2018) Glob Ecol Conserv can be useful.

Response 2: We followed Reviewer's 1 indication and introduced both sides of the argument on whether Novel Ecosystems are useful or not. (**p. 3, lines 94-105**)

Point 3: One very minor issue is that socioecological is written in three different ways along the text.

Response 3: We corrected all variations of the spelling to **social-ecological systems** (most commonly used form).

Response to Reviewer 2

We are very grateful for the detailed and thorough review and analysis that Review 2 made on our manuscript. Not only did he/she indicate the problems, but also suggested paths to address these problems. Reviewer's 2 suggestions and comments were all addressed and helped improve substantially our paper. The reviews are divided in three overarching concerns (Part 1) and further detailed comments (Part 2).

Part 1 - Overarching concerns

Overarching concern 1: A more clearly articulated research question (& associated research gap) would better motivate how the paper contributes to advancing science. The authors could go another step backwards to identify the research gap that the questions fill. Some ideas: **(a)** The following two sentences could be developed into a paragraph with some more details and references: "Understanding how societies used natural resources in the past, producing landscapes imprinted with our history, has become a very important field of scientific inquiry in recent decades. Without having a good knowledge about how the landscape transformation process took place, one cannot make good management decisions regarding its current use, resource management and biodiversity conservation." (E.g., what is considered "good knowledge" or "good management", why do we have poor knowledge or management, what are practical or scientific gaps or hurdles (including recent references that say this is a gap) or what could this knowledge achieve on the ground for sustainability, land use, or marginalized populations...?), AND/OR: **(b)** Is there any problem/drama/disagreement regarding NOVEL ecosystems and conservation?, AND/OR: **(c)** Do the authors want to say that their approach drawing on Historical Ecology, Environmental History and Historical Geography to make a historically grounded SES analysis is the novelty, then they should perhaps write 2-3 sentences here to give a brief overview of other historical SES studies and what they did or didn't achieve. (E.g., while aspect x and y have been considerably studied, no one looked at aspect z.), AND/OR: **(d)** Biocultural diversity is briefly mentioned in section 3.6. IN CASE authors want the results to speak to ongoing debates on biocultural diversity/biocultural landscapes here are some recent studies that might formulate related research gaps: Hanspach, J., et al (2020). Biocultural approaches to sustainability: A systematic review of the scientific literature.

Response: We fully addressed this concern by reframing and fleshing out our research questions (and research gap) and research goals (**p.4 lines 154-176**). **(a)** We corrected and improved the discussion on **p. 2 lines 20-35**. **(b)** We addressed this issue and introduced both sides of the argument on whether Novel Ecosystems are useful or not. (**p. 3, lines 94-105**). **(c)** We addressed this point on **p. 3 line 70-77**, the last sentence making it clear what research gap we are filling. We did not find any other historical SES studies to reference and discuss what they did or didn't achieve. **(d)** We used the reference and other biocultural diversity approach papers to connect biocultural diversity to novel ecosystems and social-ecological systems on **p. 3-4 lines 106-118**.

Overarching concern 2: Some terms could be better defined/explained for an international/interdisciplinary audience. The authors use some terms without explaining or defining them, including "resilience", "biomass resilience", "geographical tableau", "ecosystem services" (not sure if this one is considered self-explanatory these days), "macro vestiges", etc. Some of these might stem from the translation from Portuguese. To help the reader (and save them from having to google unfamiliar terms) and for the paper to reach its full potential of international and interdisciplinary spread, I would

recommend defining (even if only with a few words) the technical terms necessary for the paper's argument and using simpler words for those that are not. See detailed comments for examples.

Response: We fully addressed this concern by providing specific definitions for these terms: “resilience” (p. 2 footnote 1), “biomass resilience” (p. 9 footnote 9), “geographical tableau” (replaced with ‘geographical frame’ and explained on p. 7 line 288] “ecosystem services” (p. 5 footnote 2), “macro vestiges” (defined as apparent aspects of material culture, on p. 9 line 375).

Overarching concern 3: a) I am not fully convinced by how the paper employs the term sustainability. My third critique regards the structure and focus of Sections 3.5-3.6: Why is it important to discuss (and emphasize as a key result in the abstract) whether the historical coal production practices were ecologically, socially, and economically SUSTAINABLE? It seems unfair to judge past practices against a modern sustainability lens – and it is not clear from the paper's framing what it is supposed to achieve. (It gives the impression that Section 3.5-3.6 were written to answer a research question about sustainability, while such a question was never posed.); **b)** Regarding the “social unsustainability” of the workers' conditions and marginalized position, I also find this very relevant, although I am not convinced about the cause and effect: do the authors mean that they became MORE marginalized through the coal work?; **c)** Could the authors possibly make the link/significance of the historical findings to present-day people/biocultural diversity/conservation/land use more explicit?

Response: a) We fully agree with the reviewer and understand that the sustainability discussion does not fit in our paper. Therefore, we removed altogether the discussion regarding sustainability. We replaced it with discussions in terms of social invisibility, forest recovery and economical aspects of the charcoal production (lines 451-487). **b)** We improved the discussion of marginalization throughout the manuscript to make it more clear and added a note (p. 7 footnote 4) explaining that: “The charcoal workers, already coming from a marginalized social condition (as ex-enslaved people), became even more invisible as their work was done in a remote setting, hidden under forest cover. Charcoal work was considered as one of the socially lowest type of work, due to its rustic and dangerous nature, with dark coal staining their clothes and skin. This added to their already marginalized condition”. **c)** We made the possible links on section 3.6. (Current social-ecological dynamics; p. 511-550) and on the takeaways from the social-ecological analysis (as suggested by the reviewer); p. 11, line 504-507.

Part 2 - Detailed comments – aimed at further increasing clarity of the paper

Point 1: ABSTRACT 1-2 introductory sentences would help to understand up front what research gap the paper wishes to fill, e.g. starting with: “Historical ecology has been an important tool in deciphering human-environment interactions imprinted on landscapes throughout time, BUT it has been little applied to.../ there is lack of knowledge on...”

Response 1: We addressed this point on line 3-5.

Point 2: INTRODUCTION - Social-Ecological Systems/social-ecological system/Social-Ecological system/SES – should be consistent (I agree writing it out is the more elegant choice and easier for the reader).

Response 2: We corrected all variations of the spelling to **social-ecological systems** (most commonly used form).

Point 3: “adaptive landscape” – possibly confusing since the term adaptive can have different meanings across ecology, SES and the more socially-oriented sustainability science. The same is true for terms like “individuals” and “communities” (I spotted the latter used in two different meanings in the article), which might call for specifications like “plant and animal communities” (or what is most suitable) to avoid confusion. AND resilience (would be important to know whether the authors are using it in line with ecological or social-ecological literature)!

Response 3: We addressed and corrected the use of these terms on **p.2 line 6-8**. A specific definition for resilience (within social-ecological literature) was provided on **p. 2 footnote 1**.

Point 4: “Social Ecological Systems (SES) has been the main theoretical and methodological framework used in the emerging field of Sustainability Science” – can be debated! I would at least say “ONE OF the most important” and support it with a reference – such as the recent Clark and Harley (see eg its Table 1). There are different versions of the framework and many of the more social science-oriented Sustainability scholars are critical to this “biosphere based” Sustainability Science (they would also strongly object to the ensuing claim that SES as an important part of “social research” today – most social research still does not even consider the environment!)

Response 4: We addressed this point on **p. 2 line 16** and **line 26**.

Point 5: “good knowledge” – sounds simplistic, good for what?

Response 5: We addressed this point on **p. 2 line 34**.

Point 6: The role of the few paragraphs on pages 3-4 that talk about the three history-related fields (Historical Ecology, Environmental History and Historical Geography) is not fully clear. The authors use a language of “highlighting contributions” of fields that have (similarly to SES) “contributed to the construction of an approach that seeks to overcome the separation between nature and society”. They could take a more active stance here (do you draw on all three in the paper? Say so – and how! Is there a lack of integration between these fields and SES science? Say so – backed up by recent references! If they are merely “conceptual tools” the authors draw on, but do not help formulating the research gaps/questions, perhaps consider putting them in a separate “Conceptual framework” section (where it also might be relevant to define terms like “resilience” and “biomass resilience” and how they are linked).

Response 6: We addressed this point on **p. 3 line 70-77**, the last sentence making it clear what research gap we are filling. We believe that the sub-divisions inserted in the first section (1.1. Social-ecological systems; 1.2. Historical Coupled Human-Environment Investigation; 1.3. Novel Ecosystems and Biocultural Diversity; 1.4. Human transformation of the Atlantic Forest) will help navigate the reader through the different disciplines and conceptual tools used in the paper that help formulating the research gap and questions.

Point 7: Vestiges – is this a specific geography term? Would it be possible to use a simpler term throughout (such as traces?) – or else perhaps provide a definition and reference.

Response 7: We substituted vestiges for traces on **p. 4 line 128**.

Point 8: METHODS AND MATERIALS - Quilombos/Quilombola – the term first appears in methods section without definition. Possibly lift the explanation here from the later footnote, since not many non-Brazilians know about them and it is a recurrent & interesting feature of the paper.

Response 8: We provided a clear definition for *quilombola* on **p. 5 footnote 3**.

Point 9: “This paper synthesizes past studies conducted by two of the authors, in the last decade” – could already here add the references mentioned in the author-supplied statements (34, 39, 40, 41, 42, 43, and 44) and perhaps mention that they were published in Portuguese, which helps motivate the publication of this paper (=to reach a new audience).

Response 9: We included the reviewer’s suggestion and mentioned that they were published in Portuguese, on **p. 5 lines 211-212**.

Point 10: “historical, archaeological, vegetation and spatial data” – could be vegetational for consistent form

Response 10: Corrected to vegetational

Point 11: “iconography”? Could the authors use a few words to explain what this means as a research method.

Response 11: definition of iconography was provided on **p. 6 lines 222-223**.

Point 12: check wording and logic of this extract – “intense” lack and “clear” focus sound a bit off, and it reads like the present study IS charcoal production (rather than IS ON/ABOUT): “In the present study – charcoal production in the forests of Southeast Brazil – the lack of documental information is intense. For the charcoal manufacturing process, generic references are found, with a clear focus on metallurgy from 16th to 19th century, such as the work of Landgraf et al (37), or from the beginning of the 20th century (38).”

Response 12: We corrected these sentences on **p. 6 lines 226-229**.

Point 13: “Concerning sampling techniques” – confusing, I find that the paragraph does not explain about sampling techniques. What is the role of two different study areas, do their different features allow for some kind of comparison? (“also” in the next sentence is confusing – it reads as if the field methods are another sampling technique, is this the intention?)

Response 13: We corrected the sentence and explained the role of the two study areas on **p. 6 lines 231-234**.

Point 14: “kiln” – could this term possibly be explained at first mention – e.g. by oven/production site or what would be the proximate word – and/or by referring to the excellent Figure 1?

Response 14: We referred kiln to Figure 1 as suggested.

Point 15: “first-level” “second-tier” – Between this explanation, the result section, and Figure 3, I am a bit confused by the notion first/second (level)/tier (choose one?!) variable. First/second in relation to what scale? Adding a sentence here, and/or redesigning Figure 3 to elucidate this, might be useful to explain/exemplify the levels. I also wonder why some, but not all, of these abbreviations (e.g. U and ECO) show up in Table 3 and Figure 7.

Response 15: We corrected this part and only used tier and not level. We also improved the explanation at the end of the paragraph on **p. 6 lines 264-266**.

Point 16: RESULTS/Sections 3.1 - 3.4 - "There is a limited amount of written documentation about charcoal production prior to the 20th century THAT CAN BE FOUND." – last words are redundant.

Response 16: This was corrected.

Point 17: "Inscribed in the current landscape as signs or traces of different NATURES" – I suppose the authors refer to different "types/kinds"? Perhaps avoid using the term nature in other meanings than "natural environment" to avoid confusing the reader ("such as" before the colon can also be dropped to avoid repetition).

Response 17: This was corrected.

Point 18: "the charcoal activity demanded very few initial inputs (of tools) and guaranteed relative independence" – I understand there is little documentation, but do the authors have any idea or estimated guess as to who bought the coal (if people in the city, how it was transported from the remote areas?) and what was offered as payment (money/goods?). Would be relevant to know to understand how it contributed to their subsistence.

Response 18: Although relevant, we do not have the proper information to answer this question. We feel that while it would be interesting to have this answer, it does not compromise the understanding of this section of the paper.

Point 19: "geographical tableau" – not even with the help of Google I could find an explanation for this term (in English). Can it be explained/defined, even if only with a few words? Following the reference, I get to Gomes' concept of "quadros geográficos" – would perhaps be interesting to lift this into the text for the international reader, sth like "Borrowing a concept from Brazilian Geographer Paulo da Costa Gomes, such evidence constitutes today a 'geographical frame' (quadro geográfico), meaning.../referring to [geography as a graphic way of structuring thought]. This frame allows us to think...." (I prefer frame over tableau since it has the same double meaning of theoretical frame(work)/lens and painting, see also <https://journals.openedition.org/confins/21686>)

Response 19: "geographical tableau" was replaced with '**geographical frame**' and explained on **p. 7 line 288**.

Point 20: "the worker's management planning" – do you mean workers' (plural)?

Response 20: Corrected to **workers'** (plural)

Point 21: "Our research is still under development"???? – confusing, will you add more results, or do you mean that the paper is the first of a number of planned publications?

Response 21: We removed this opening phrase to avoid confusion, on **p. 7 line 330**.

Point 22: "In the Pedra Branca forest, 104 ruins were found and 107 ruins found in the Tijuca Massif." – confusing sentence structure, using the same format/order for both would help the reader.

Response 22: We corrected this sentence by using the same format/order to help with clarity on **p. 7 line 330**.

Point 23: “In the Pedra Branca Massif, although the vast majority of charcoal sites and ruins are located below 300 m altitude, only 30% are above this elevation.” – delete “although”, I see no opposition between the first and second statement (could be joined with semicolon) – whole paragraph (about the elevation) could be revised for consistency and clarity.

Response 23: We deleted “although” as suggested.

Point 24: “steep, elevated terrain and distant from lowland areas” – seems to be a noun missing after elevated?

Response 24: We addressed this point on **p. 8 line 348**.

Point 25: “it was a parallel and marginal occupation for the farmers and land owners”. – I suppose you mean “TO/(in the eyes of) the farmers and land owners, it was a parallel and marginal occupation.” – “for” sounds like they were the ones carrying it out.

Response 25: We corrected this as suggested.

Point 26: “In part, this is due to a fact that may have played a role” confusing sentence, either it is a fact or a possibility, can be put more simply as → “The regrowth of stumps may have played a role in the return of the forest: some species that were cut for charcoal production are able to regrow and, since they lose apical dominance, they created bifurcations in the regrowth process (Figure 5) (60, 65).

Response 26: We corrected this sentence as suggested on **p. 9 lines 369-372**.

Point 27: “constitutes a macro vestige of past use of the forest” – what is a macro vestige? Can you use a simpler term? Or just delete the clause to avoid repetition with next sentence: ☐ “This pattern is relatively common in areas that were deforested. The (!) resprouted tree stumps compose one of the vegetation indicators of past forest use alongside native species with anomalous distributions and persistence and dominance of previously introduced exotic species, especially fruit trees and ritualistic herbs (60).”

Response 27: We chose to define ‘macro vestige’ (visible aspects of material culture) on **p.9 lines 376-377**.

Point 28: “and that was also consumed by the charcoal workers” - do you mean “it”?

Response 28: We corrected to “it” as suggested.

Point 29: “Jackfruit is rich in carbohydrate and protein and of low cost, it was potentially transported to the areas of its work in the forest, germinating from the uneaten remains and discarded in the forest.” – Check this sentence for repetition (low cost) and logic (should it be “their work” and “remains discarded”?) – check the rest of the paragraph for grammar/punctuation (e.g. “ON the edge”)

Response 29: We addressed these issues on **p. 9 lines 382-384**.

Point 30: *Carioca* – word needs explanation to non-Portuguese speakers.

Response 30: We explained *Carioca* on **p. 9 footnote 20**.

Point 31: Sections 3.5 - 3.6 (The first paragraph in 3.5 sounds a bit more like methods language to me.)

Response 31: We justify the permanence of the first paragraph in this section and not in the methods section as it the part where the framework is presented and used. A second option, which to us does not seem the best, would be to move this paragraph up to the introduction, where the "bases" of the article are presented.

Point 32: It could be clearer in this section what are the main takeaways from the impressive Table 3. (That address research gaps/contribute to new knowledge).

Response 32: We included a paragraph with the main takeaways at the end of the section on **p. 10 lines 501-808**.

Point 33: "biomass resilience"? term was never defined – does it relate to the SES framework or is it an ecological term?

Response 33: We provided a definition for "biomass resilience" on **p. 9 footnote 9**.

Point 34: "Ecological novelty includes never before seen patterns of species composition and relative abundance in a given biome (22), that include anomalous distribution of native species, which may present super dominant populations." – repetition of "include"

Response 34: We corrected this issue on **p.9 line 409**.

Point 35: "energetic sustainability" – do you mean energy supply? Why the need to call it sustainability

Response 35: We corrected to "energy supply" on **p. 10 line 426**.

Point 36 "with increase (enforcement of) forest protection legislation at the same time that the demand for charcoal increased" – do you mean increased? (possibly use other word to avoid repetition, "as FPL was rolled out/enforced/... at the same time that")

Response 36: We corrected this on **p. 10 line 451**.

Point 37: "Patzlaff (50) ... identified tree species that were burned in the past" – I would be curious of how many he identified and perhaps which were most common?

Response 37: We chose to leave the text as it is, not extracting more in depth vegetation composition research results from Patzlaff (63).

Point 38: "The ecosystem has kept marks of this history in many of its attributes, SELECTING long living early-secondary native and exotic species" – do you mean "including"/"such as"?

Response 38: We replaced "selecting" with "such as" on **p. 10 line 473**.

Point 39: CONCLUSION "lighter" – is this the right word, would torch be better?

Response 39: we replaced "lighter" with "fire" on **p. 12 line 566**.

Point 40: "at the cost of their own invisibility and marginalization" – does this mean they become MORE marginalized???? Not sure this is currently backed up.

Response 40: This is better explained on **p. 7 footnote 4** explaining that: "The charcoal workers, already coming from a marginalized social condition (as ex-enlaved people), became even more invisible as

their work was done in a remote setting, hidden under forest cover. Charcoal work was considered as one of the socially lowest type of work, due to its rustic and dangerous nature, with dark coal staining their clothes and skin. This added to their already marginalized condition”.

Point 41: “Reforestation efforts in the north facing slopes ... is further hindered” – should be “are”.

Response 41: We corrected this point.

Point 41: “In terms of sustainability it would be far more advantageous to harvest the jackfruit from these novel ecosystems, generating income and dealing with issues of food insecurity in the slums of Rio” – I agree (with the point and the use of sustainability)!!! this would be an interesting research project! (also corresponding with the ‘fitness’/organic/vegetarian trend we describe in our study on biocultural diversity in Rocinha) – although there is a conflict with a latter sentence mentioning jackfruit as an “important source of food for the population” – this makes it seem like this is already the case?

Response 41: We increased the discussion on jackfruit on **section 3.6.** (Current social-ecological dynamics) since we incorporated the use of biocultural diversity concept, as one of the suggestions of the reviewer. We clarified the second sentence “important source of food for the population” on **p. 13 lines 630-633.**

Point 42: Argumentation in this last (long!) sentence is somewhat circular: “Although some management issues remain to be resolved, such as the management of exotic fruit species that still provides an important source of food for the population (both marginalized and affluent), it is noteworthy the efforts of flexible (and perhaps not purposeful) management of the Pedra Branca state park to incorporate quilombo areas and areas of persimmon and banana production, important forest products, within its limits, albeit with some territorial conflicts unsolved.” Is it really a concluding remark, it sounds more like new topics are being introduced (which might be more relevant to bring up in sections 3.5-3.6)?

Response 42: We improved the sentence and removed this discussion from conclusions and placed it on **section 3.6.** (Current social-ecological dynamics) **p.11 line 524-529**, and improved the final concluding remarks on **p. 13 lines 628-633.**

Point 43: TABLES AND FIGURES The tables and figures are great and really help to illustrate the text! Table 3 is impressive and gives an overview of the complex interactions! It is however not so intuitive from the table why some variables appear horizontally and others vertically (i.e. 1 tier, 2 tier). Perhaps including such information in the titles (or possibly using another layout) could help to see this? (Also check for consistent use of small/capital letters and hyphen/n-dash).

Response 43: We addressed these issues fully explaining why some variables appear horizontally and other vertically on the Figure and table captions section, on **p. 19 Table 3** and also checked for consistent use of small/capital letters and hyphen/n-dash.

Point 44: If short of space for figures, Fig 3-4 could be combined into one (might also allow for better comparison).

Response 44: If possible, we would prefer to maintain figure 3 and 4 separated for better visualization.

Point 45: Figure 7: Great figure! Would it be possible to repeat the abbreviations in the caption?

Response 45: We repeated the abbreviation in the caption.